# Distributed Machine Learning with Sparse Heterogeneous Data

**Dominic Richards**
Department of Statistics
University of Oxford
24-29 St Giles', Oxford, OX1 3LB
`Dominic.Richards94@gmail.com`

**Sahand N Negahban**
Department of Statistics and Data Science
Yale University
24 Hillhouse Ave., New Haven, CT 06510
`Sahand.Negahban@Yale.edu`

**Patrick Rebeschini**
Department of Statistics
University of Oxford
24-29 St Giles', Oxford, OX1 3LB
`Patrick.Rebeschini@stats.ox.ac.uk`

## Abstract

Motivated by distributed machine learning settings such as Federated Learning, we consider the problem of fitting a statistical model across a distributed collection of heterogeneous data sets whose similarity structure is encoded by a graph topology. Precisely, we analyse the case where each node is associated with fitting a sparse linear model, and edges join two nodes if the difference of their solutions is also sparse. We propose a method based on Basis Pursuit Denoising with a total variation penalty, and provide finite sample guarantees for sub-Gaussian design matrices. Taking the root of the tree as a reference node, we show that if the sparsity of the differences across nodes is smaller than the sparsity at the root, then recovery is successful with fewer samples than by solving the problems independently, or by using methods that rely on a large overlap in the signal supports, such as the group Lasso. We consider both the noiseless and noisy setting, and numerically investigate the performance of distributed methods based on Distributed Alternating Direction Methods of Multipliers (ADMM) and hyperspectral unmixing.

## 1 Introduction

The wide adoption of mobile phones, wearable and smart devices, has created a influx of data which requires processing and storage. Due to the size of these datasets and communication limitations, it is then often not feasible to send all the data to a single computer cluster for storage and processing. This has motivated the adoption of decentralised methods, such as Federated Learning [33, 43], which involves both storing and processing data locally on edge devices.

This increase in data sources has led to applications that are increasingly high-dimensional. To be both statistically and computationally efficient in this setting, it is then important to develop approaches that can exploit the structure within the data. A natural assumption in this case is that the data is sparse in some sense. For instance, a subset of features is assumed to be responsible for determining the outcome of interest or, in the case of compressed sensing [13], the data is assumed to be generated from a sparse signal.

Fitting statistical models on data collected and stored across a variety of devices presents a number of computational and statistical challenges. Specifically, the distributed data sets can be heterogeneous

35th Conference on Neural Information Processing Systems (NeurIPS 2021).

owing to data coming from different population distributions e.g. each device can have different geographic locations, specifications and users. Meanwhile, from a computational perspective, it is often unfeasible, due to network limitations and robustness, to have a single central "master" device collect and disseminate information. This creates a demand for statistical methodologies which are both: flexible enough to model potential statistical differences in the underlying data; and can be fit in a decentralised manner i.e. without the need for a centralised device to collate information.

In this work, we set to investigate the statistical efficiency of a method for jointly fitting a collection of *sparse* models across a collection of heterogeneous datasets. More precisely, models are associated to nodes within a graph, with edges then joining two models if their *difference* is also assumed to be *sparse*. The approach of penalising the differences between models (in an appropriate norm) has, for instance, been applied within both Federated Learning [28] and Hyperspectral Denoising [23, 14] to encode heterogeneous data. In our case, we consider linear models and obtain precise insights into when jointly fitting models across heterogeneous datasets yield gains in statistical efficiency over other methods, such as, the group Lasso and Dirty Model for multi-task learning [24]. In addition to being solvable in a centralised manner with standard optimisation packages, the framework we consider is then directly applicable to decentralised applications, as information only needs to be communicated across nodes/models/devices which share an edge within the graph.

## 1.1 Our Contribution

We consider a total variation scheme that penalises differences between models that share an edge in the graph. This scheme encodes the intuition that if the signal differences are sufficiently sparse then, to recover all signals in the graph, it is more statistically efficient to first recover a single signal associated to a particular reference node (root) and then recover the signal differences associated to edges. Following the celebrated Basis Pursuit algorithm [5], we consider the solution that minimises the $\ell_1$ norm of the model associated to a root node of the tree and the differences between models that share an edge. We refer to this method as *Total Variation Basis Pursuit (TVBP)*. A noisy variant similar to Basis Pursuit Denoising [9] is then considered, where the linear constraint is substituted for a bound on the $\ell_2$ norm of the residuals. We call this method as *Total Variation Basis Pursuit Denoising (TVBPD)*. Note that variants of TVBPD have been successfully applied within the context of hyperspectral data [23, 14] and thus, this work now provides theoretical justification in this case.

Given this framework and assuming sub-Gaussian design matrices, we show that statistical savings can be achieved by TVBP and TVBPD as opposed to solving them either independently or with methods that consider the union of supports (see Table 1). In the noiseless case, TVBP requires a total sample complexity of $O(s + n^2 s')$ where $s$ is the root sparsity, $s'$ is the maximum sparsity of the difference along edges and $n$ is the number of nodes. In contrast, methods like the group Lasso and Dirty Model [24] have an $O(ns)$ total sample complexity, which matches the case when nodes do not communicate with each other. Moreover, note that the TVBP approach does *not* need to know the true underlying graph $G$, whereas the optimal stepwise approach does. If the true graph $G$ is known, TVBP can incorporate this information to yield improved sampled complexity (see Table 2 in Section 2.2). In the noisy setting, we demonstrate that TVBPD has an $\ell_1$ estimation error bounded as $O(\sqrt{s} + \sqrt{ns'})$, where as the stepwise approach scales as $O(s + n\sqrt{s'})$, and thus, achieves an $O(\sqrt{ns'})$ saving in this case. While the step wise approach achieves optimal total sample complexity in the noiseless case, in the noisy setting its estimation scales sub-optimally compared to TVBPD.

Alongside total sample complexity savings over other joint recovery methods, we also show that TVBP is amenable to decentralised machine learning. Specifically, the objective can be reformulated with constraints that reflect the graph topology allowing a Distributed Alternating Direction Methods of Multipliers (ADMM) algorithm [3] to be applied. Theoretical findings are also supported by experiments (Section 3.2) which demonstrate both: TVBP can outperform group Lasso methods [24] when model differences are sparse; and TVBPD yields qualitative improvements in hyperspectral unmixing with the real-world AVIRIS Cuprite mine data set.

A key reason for TVBP achieving sample complexity savings over variants of the group Lasso and Dirty Model [24], is that the matrices associated to each tasks *do not* need to satisfy an *incoherence condition* with respect to their support sets. The incoherence condition is a common assumption within the analysis of sparse methods like the Lasso (see for instance [49]) and, in short, it requires the empirical covariance of the design matrix to be invertable when restricted to co-ordinates in the signal support set. This immediately implies that the sample size at each task is larger than the support

| Method | Total Sample Complexity $\sum_{v \in V} N_v$ | Sim. Recovery | Know $G$ |
|---|---|---|---|
| Independent Basis Pursuit | $ns + \text{Diam}(G)^2 s'$ | ✓ | ✗ |
| Stepwise Basis Pursuit | $s + ns'$ | ✗ | ✓ |
| Group Lasso [36, 37] Dirty Model [24] GSP [16] | $ns + \text{Diam}(G)^2 s'$ | ✓ | ✗ |
| TVBP (this work) | $s + n^2\text{Diam}(G)s'$ | ✓ | ✗ |

Table 1: Worst case total sample complexities (up-to logarithmic and constant factors) for recovering a collection of sparse signals $\{x_v^\star\}_{v \in V}$ on a tree graph $G = (V, E)$ in the noiseless case with sub-Gaussian design matrices. Sparsity of the root signal $|\text{Supp}(x_1^\star)| \leq s$, sparsity of difference along edges $e = (v, w) \in E$ $|\text{Supp}(x_v^\star - x_w^\star)| \leq s'$. *Sim. Recovery*: Whether the method simultaneously recovers the collection of signals $\{x_v^\star\}_{v \in V}$. *Know $G$*: Whether the sample complexity listed requires knowledge of relationship graph $G$. *TVBP*: Total Variation Basis Pursuit (1). When the algorithm *does not* depend on $G$ and $s'$, the best case of $G, s'$ can taken.

set of that given task. In contrast, for TVBP an alternative analysis is conducted by reformulating the problem into a standard basis pursuit objective with an augmented matrix and support set. In this case, the problem structure can be leveraged to show a *Restricted Null Space Property* holds when the sample size at each task scales with the sparsity along the graph edges. This highlights that the Total Variation penalty encodes a different structure when compared to methods like the group Lasso and Dirty Model [24]. As shown in the noisy case, this approach can then be generalised through the *Robust Null Space Property* [8, 17].

## 1.2 Related Literature

Learning from a collection of heterogeneous datasets can be framed as an instance of multi-task learning [6], with applications in distributed contexts gaining increased attention recently. We highlight the most relevant to our setting. The works [52, 48] have considered models penalised in an $\ell_2$ sense according to the network topology to encode prior information. The $\ell_2$ penalty is not appropriate for the sparse setting of our work. A number of distributed algorithms have been developed for the sparse setting, for a full review we refer to [1]. The works [24, 44, 50, 29, 51, 38] have developed distributed algorithms following the group Lasso setting, in that, the signals are assumed to be composed of a common shared component plus an individual component. Within [24, 44, 50] this requires each node to satisfy an incoherence condition, while the setting in [29, 38] is a specific case of a star topology within our work. For details on how the incoherence condition influences the sample complexity see discussion in Section 2.1. The work [40] develops a manifold lifting algorithm to jointly recover signals in the absence of an incoherence assumption, although no theoretical guarantees are given.

Federated machine learning [27, 43, 33, 28] is a particular instance of distributed multi-task learning where a central node (root) holds a global model and other devices collect data and update their model with the root. Data heterogeneity can negatively impact the performance of methods which assume homogeneous data, see [33] for the case of Federated Averaging. This motivates modelling the data heterogeneity, as recently done within [28] where the difference in model parameters is penalised at each step. Our work follows this approach in the case of learning sparse models, and thus, it provides insights into when an improvement in statistical performance can be achieved over other baseline methods.

Simultaneously recovering a collection of sparse vectors can also be framed into the multiple measurement vectors framework [15], which has been precisely investigated for $\ell_1/\ell_q$ regularisation for $q > 1$. Specifically, $\ell_1/\ell_\infty$ was investigated within [55, 36, 47] and $\ell_1/\ell_2$ in [31, 37]. Other variants include the Dirty Model of [24], multi-level Lasso of [32] and tree-guided graph Lasso of [25]. In the same context, a number of works have investigated variants of greedy pursuit style algorithms [16, 11, 12, 46]. These methods assume a large overlap between the signals, with their analysis often assuming each task satisfies an incoherence condition [36, 24, 37] (see Section 2.1).

The total variation penalty is linked with the fused Lasso [53, 21, 45, 7, 42] and has been widely applied to images due to it promoting piece-wise continuous signals which avoids blurring. As

far as we are aware, the only work theoretically investigating the total variation penalty as a tool to link a collection of sparse linear recovery problems has been [10]. This work considers the penalised noisy setting and gives both asymptotic statistical guarantees and an optimisation algorithm targeting a smoothed objective. In contrast, we give finite sample guarantees as well as settings where statistical savings are achieved. The application of hyperspectral unmixing [22, 23, 14] has successfully integrated the total variation penalty within their analysis. Here, each pixel in an image can be associated to its own sparse recovery problem, for instance, the presence of minerals [23] or the ground class e.g. trees, meadows etc. [14].

## 2 Noiseless Setting

This section presents results for the noiseless setting. Section 2.1 introduces the setup and details behind the comparison to other methods in Table 1. Section 2.2 presents analysis for *Total Variation Basis Pursuit* alongside descriptions of how refined bounds can be achieved. Section 2.3 presents experimental results for the noiseless setting. Section 2.4 gives a sketch proof of the main theorem.

### 2.1 Setup

Consider an undirected graph $G = (V, E)$ with nodes $|V| = n$ and edges $E \subseteq V \times V$. Denote the degree of a node $v \in V$ by $\text{Deg}(v) = |\{(i, j) \in E : i = v \text{ or } j = v\}|$ and index the nodes $V = \{1, \ldots, n\}$ with a root node associated to 1. To each node $v \in V$, associate a signal vector $x_v^\star \in \mathbb{R}^d$. The objective is to estimate the signals $\{x_v^\star\}_{v \in V}$ through measurements $\{y_v\}_{v \in V}$ defined as $y_v = A_v x_v^\star \in \mathbb{R}^{N_v}$ where $A_v \in \mathbb{R}^{N_v \times d}$ is a design matrix. As we now go on to describe, we will assume the signals are both: sparse $x_v^\star$ for $v \in V$, and related through the graph $G$. For instance, the graph $G$ can encode a collection of wearable devices connected through a network. Each node holds a collection of data $(y_v, A_v)$ that can, for example, represent some sensor outputs. Alternatively, in Hyperspectral Denoising, each node $v \in V$ is associated to a pixel in a image and the signal $x_v^\star$ indicates the presence of a mineral or classification of land type. The graph can then encode the landscape topology.

Following these examples, it is natural that the collection of signals $\{x_v^\star\}_{v \in V}$ will have a sparsity structure related to the graph $G$. Specifically, if two nodes share an edge $e = (v, w) \in E$ then it is reasonable that only a few co-ordinates will change from $x_v^\star$ to $x_w^\star$. For instance, in Hyperspectral Imaging we expect the composition of the ground to change by a few minerals when moving to an adjacent pixel. This can then be explicitly encoded by assuming difference in the underlying signals can also be sparse. We encode the structural assumption on the signal within the following definition.

**Definition 1** (($G, s, s'$) Sparsity)**.** *A collection of signals $\{x_v^\star\}_{v \in V}$ is $(G, s, s')$-sparse if the following is satisfied. The root-node signal support is bounded $|Supp(x_1^\star)| \leq s$. For any edge $e = (v, w) \in E$ the support of the difference is bounded $|Supp(x_v - x_w)| \leq s'$.*

We are interested in the total number of samples $N_{\text{Total Samples}} := \sum_{v \in V} N_v$ required to recover all of the signals $\{x_v^\star\}_{v \in V}$. We begin by describing the total number of samples $N_{\text{Total Samples}}$ required by baseline methods, a summary of which in Table 1.

**Independent Basis Pursuit** For an edge $e = (v, w) \in E$, denote the support of the difference as $S_e = \text{supp}(x_v^\star - x_w^\star)$. Let us suppose for any pair of edges $e, e' \in E$ the supports of the differences are disjoint from each other $S_e \cap S_{e'} = \emptyset$ and the support of the root $S_e \cap S_1 = \emptyset$. Let $G$ be a tree graph and the integer $i_v \in \{0, \ldots, n-1\}$ denote the graph distance from node $v$ to the root agent 1. If each node has sub-Gaussian matrices $A_v$ and performed Basis Pursuit independently, then the number of samples required by agent $v$ to recover $x_v^\star$ scales as $N_v \geq s + i_v s'$. The total sample complexity is at least $N_{\text{Total Samples}} = \sum_{v \in V} N_v \geq ns + s' \sum_{v \in V} i_v = O(ns + \text{Diam}(G)^2 s')$ where we lower bound $\sum_{v \in V} i_v$ by considering the longest path in the graph including agent 1. The optimisation problem associated to this approach is then

$$\min_{x_v} \|x_v\|_1 \text{ subject to } A_v x_v = y_v \text{ for } v \in V,$$

which can be solved independently for each agent $v \in V$.

**Stepwise Basis Pursuit** Consider the support set structure described in **Independent Basis Pursuit**. The signals can then be recovered in a stepwise manner with a total sample complexity of $O(s + ns')$.

Precisely, order $s$ samples can recover the root signal $x_1^\star$, meanwhile order $n \times s'$ samples can recover each of the differences associated to the edges. Any node's signal can then be recovered by summing up the differences along the edges. This yields a saving from $O(ns + n^2 s')$ to $O(s + ns')$, which is significant when the difference sparsity $s'$ is small and the network size $n$ is large. This embodies the main intuition for the statistical savings that we set to unveil in our work. The associated optimisation problem in this case requires multiple steps. Let the solution at the root be denoted $\widehat{x}_1 = \arg\min_{x_1 : A_1 x_1 = y_1} \|x_1\|_1$. For a path graph we recursively solve, with the notation $\sum_{i=2}^{1} \cdot = 0$, the following sequence of optimisation problems for $j = 2, \ldots, n$

$$\min_{\Delta} \|\Delta\|_1 \text{ subject to } A_j \Delta = y_j - A_j \widehat{x}_1 - \sum_{i=2}^{j-1} A_k \widehat{\Delta}_i,$$

with the sequence of solutions here denoted $\{\widehat{\Delta}_j\}_{j=2}^n$.

**Group Lasso / Dirty Model / GSP** The group Lasso [36, 37], Dirty Model [24] and Greedy Pursuit Style algorithms [16, 11, 12, 46] from the Multiple Measurement Vector Framework [15], assume an incoherence condition on the matrices $A_v$ for $v \in V$, which impacts the total sample complexity. Namely, for $v \in V$ denote the support $S_v = \text{Supp}(x_v^\star) \subseteq \{1, \ldots, d\}$ alongside the design matrix restricted to the co-ordinates in $S_v$ by $(A_v)_{S_v} \in \mathbb{R}^{N_v \times |S_v|}$. The incoherence assumption for the Dirty Model [24] then requires $(A_v)_{S_v}^\top (A_v)_{S_v}$ to be full-rank (invertibility), and thus, $N_v \geq |S_v|$. Since $|S_v| = s + i_v s'$ the lower bound on $N_{\text{Total Samples}}$ then comes from **Independent Basis Pursuit**. One of the associated optimisation problem for the group lasso then takes the form for some $\lambda > 0$

$$\min_{x_1, x_2, \ldots, x_n} \sum_{v \in V} \|y_v - A_v x_v\|_2^2 + \lambda \sum_{j=1}^{d} \|((x_1)_j, (x_2)_j, \ldots, (x_n)_j)\|_2.$$

## 2.2 Total Variation Basis Pursuit

To simultaneously recover the signals $\{x_v^\star\}_{v \in V}$ we consider the *Total Variation Basis Pursuit* (TVBP) problem. Specifically for a tree-graph $\widetilde{G} = (V, \widetilde{E})$ with edges $\widetilde{E} \subset V \times V$, consider:

$$\min_{x_1, x_2, \ldots, x_n} \|x_1\|_1 + \sum_{e=(v,w) \in \widetilde{E}} \|x_v - x_w\|_1 \quad \text{subject to} \quad A_v x_v = y_v \text{ for } v \in V. \tag{1}$$

Let us denote a solution to (1) as $\{x_v^{TVBP}\}_{v \in V}$. Note that $\widetilde{G}$ does not have to be equal to the graph associated to the sparsity of $\{x_v^\star\}_{v \in V}$. For instance, we can consider a star graph for $\widetilde{G}$ whilst $G$ is a more complex unknown graph. Furthermore, we highlight that (1) couples the agents solutions together in a distinctly different manner when compared to the methods in Section 2.1 e.g. Group Lasso.

We now upper bound on the number of samples $N_1, N_{\text{Non-root}}$ required for TVBP to recover the signals. For the following, we say that if $A$ has independent and identically distributed (i.i.d.) sub-Gaussian entries, then the $i, j$th entry $A_{ij}$ satisfies $\mathbb{P}(|A_{ij}| \geq t) \leq \beta e^{-\kappa t^2}$ for all $t \geq 0$ for sub-Gaussian parameters $\beta$ and $\kappa$. Let us also denote the root sample size as $N_{\text{Root}} = N_1$, with all non-root agents having the same sample size $N_{\text{Non-root}} = N_v$ for $v \in V \setminus \{1\}$. The proof for the following theorem can then be found in Appendix A.1.1.

**Theorem 1.** *Suppose the signals $\{x_v^\star\}_{v \in V}$ are $(G, s, s')$-sparse and the matrices satisfy $A_v = \frac{1}{\sqrt{N_v}} \widetilde{A}_v$ where $\{\widetilde{A}_v\}_{v \in V}$ each have i.i.d. sub-Gaussian entries. Fix $\epsilon > 0$. If*

$$N_{Root} \gtrsim \max\{s, n^2 Diam(G)s'\}\big(\log(d) + \log(1/\epsilon)\big) \text{ and}$$
$$N_{Non\text{-}root} \gtrsim nDiam(G)s'\big(\log(d) + \log(n/\epsilon)\big),$$

*then with probability greater than $1 - \epsilon$ the TVBP solution with a star graph $\widetilde{G}$ is unique and satisfies $x_v^{TVBP} = x_v^\star$ for any $v \in V$.*

Theorem 1 provides conditions on the number of samples held by each agent in order for TVBP to recover the signals when $\widetilde{G}$ is a star topology. As seen in Table 1, the total number of samples in

this case satisfies $N_{\text{Total Samples}} = O(s + n^2 \text{Diam}(G)s')$. As we now go on to describe, the sample complexities in Theorem 1 are *worst case* since we have assumed no prior knowledge of $G$.

Incorporating knowledge of the signal graph $G$ into the TVBP problem (1) naturally influences the total sample complexity required for recovery. Table 2 provides a summary of the total complexity required in two different cases. Precisely, an improved total sample complexities can be achieved when: $G$ is a known tree graph, as well as when the non-root design matrices $\{A_v\}_{v \in V \setminus \{1\}}$ are the same. When $G$ is a known tree graph, the sample complexity is reduced to $s + \max\{n^2, n\text{Deg}(V \setminus \{1\})^2 \text{Diam}(G)^2\}s'$ from $s + n^2\text{Diam}(G)$ previously. In the case $\text{Diam}(G) = O(\sqrt{n})$, then an order $\sqrt{n}s'$ saving in achieved. Meanwhile, if $\{A_v\}_{v \in V \setminus \{1\}}$ are also equal, then the sample complexity reduces to $s + n\text{Deg}(1)^2 s'$ which, for constant degree root nodes $\text{Deg}(1)$, matches the optimal stepwise method. This precisely arises due to the null-spaces of the non-root nodes matrices being equal in this case, allowing the analysis to be simplified. Details of this are provided within the proof sketch. The assumption that the sensing matrices $\{A_v\}_{v \in V \setminus \{1\}}$ are equal is then natural for compressed sensing and Hyperspectral applications as $A_v$ represents the library of known spectra, and thus, can be identical across the nodes.

| Method & Assumptions | Total Sample Complexity $\sum_{v \in V} N_v$ | Know $G$ |
|---|---|---|
| TVBP | $s + n^2\text{Diam}(G)s'$ | ✗ |
| TVBP + $G$ Tree | $s + \max\{n^2, n\text{Deg}(V \setminus \{1\})^2\text{Diam}(G)^2\}s'$ | ✓ |
| TVBP + $G$ Tree + $\{A_v\}_{v \in V \setminus \{1\}}$ equal | $s + n\text{Deg}(1)^2 s'$ | ✓ |

Table 2: Setting as described in Table 1. Comparing total sample complexity for TVBP with different assumptions: whether $G$ is a known tree; or $G$ is a known tree and the design matrices $\{A_v\}_{v \in V \setminus \{1\}}$ are identical. Formal results can be found in Theorems 3 and 4 within the Appendix A.1. When the algorithm depends upon the choice of graph $G$, the rate given depends upon this choice of $G$ and the associated sparsity along edges $s'$.

## 2.3 Experiments for Noiseless Case

This section presents numerical experiments for Total Variation Basis Pursuit problem (1). The paragraph **Statistical Performance** focuses on the statistical performance of TVBP, supporting the results summarised in Table 2. Paragraph **Distributed Algorithm** outlines how the objective (1) can be solved in a decentralised manner.

**Statistical Performance** Figure 1 plots the probability of recovery against the number of samples held by non-root nodes $N_v$ for $v \in V \setminus \{1\}$ with a fixed number of root agent samples $N_1 = \lfloor 2s\log(ed/s)\rfloor$. Observe, for a path topology and balanced tree topology, once the non-root nodes have beyond approximately 30 samples, the solution to TVBP finds the correct support for all of graph sizes. In contrast, the number of samples required to recover a signal with Basis Pursuit at the same level of sparsity and dimension considered would require at least 80 samples, i.e. $2s\log(ed/s)$. We therefore save approximately 50 for each non-root problem.

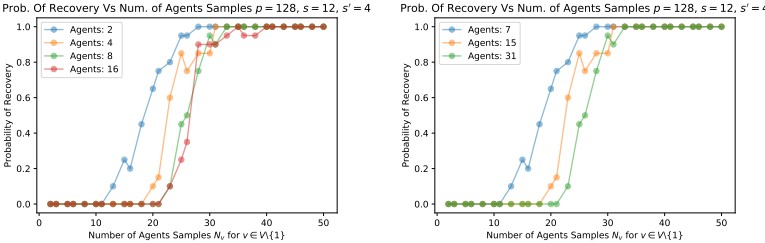

Figure 1: Probability of recovery vs number of non-root node samples $N_v$ for $v \in V \setminus \{1\}$. Problem setting $d = 128$, $s = 12$, $s' = 4$ and $N_1 = \lfloor 2s\log(ed/s)\rfloor = 80$, for path (*Left*) and balance tree with branches of size 2 (*Right*). Lines indicates graph sizes with $n \in \{2, 4, 8, 16\}$ for path and $n \in \{7, 15, 31\}$ for balanced tree with heights of $\{2, 3, 4\}$ respectively. Solution to reformulated problem (11) found using CVXOPT. Each point is an average of 20 replications. Signal sampled from $\{1, -1\}$, differences concatenation of $s'$ values. $\{A_v\}_{v \in V}$ standard Gaussian and $\widetilde{G} = G$.

**Distributed Algorithm** To solve the optimisation problem (1) in a decentralised manner an ADMM algorithm can be formulated, the details of which are given in Appendix A.2. The Optimisation Error for the method is plotted in Figure 2, which is seen to converge with a linear rate. The convergence for a path topology is slower, reaching a precision of $10^{-8}$ in 300 iterations for 7 nodes, while the same size balanced tree topology reaches a precision of $10^{-15}$. This is expected as the balanced trees considered are more connected than a path, and thus, information propagates around the nodes quicker. Larger tree topologies also require additional iterations to reach the same precision, with a size 63 tree reaching a precision of $10^{-7.5}$ in 300 iterations.

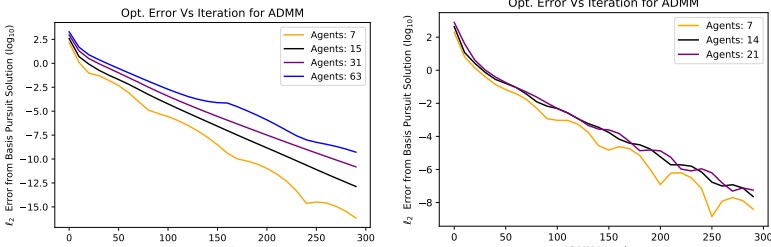

Figure 2: Optimisation error $\|x^t - x^\star_{\text{BP}}\|^2_2$ (Log scale) vs Iterations for ADMM method with $\rho = 10$ for different graph sizes (lines) and topologies (plots). Here $x^\star_{\text{BP}}$ is solution to (1). Problem parameters $d = 2^9$, $s = \lfloor 0.1d \rfloor$ and $s' = 4$. *Left*: Balanced trees, branch size 2 and heights $\{2, 3, 4, 5\}$. *Right*: Path topology. Agent sample size $N_1 = 2s \log(ep/2s)$ and $N_v = 150$ for $v \neq 1$. Matrices $\{A_v\}_{v \in V}$ i.i.d. standard Gaussian entries, $x^\star_1$ has $s$ values randomly drawn from $\{+1, -1\}$ and $\{\Delta^\star_e\}_{e \in E}$ each have $s'$ i.i.d. standard Gaussian entries, locations chosen at random.

## 2.4 Proof Sketch for Theorem 1

This section provides a proof sketch for Theorem 1. For a response $y \in \mathbb{R}^N$ and design matrix $A \in \mathbb{R}^{N \times d}$, recall that the *Basis Pursuit* problem is given by

$$\min \|x\|_1 \text{ subject to } Ax = y. \tag{2}$$

Let us denote the solution to (2) as $x^{\text{BP}}$. It is known that if $y = Ax^\star$ for some sparse vector $x^\star$ supported on $S$, then the solution to (2) recovers the signal $x^{\text{BP}} = x^\star$ if and only if $A$ satisfies the *Restricted Null Space Property* with respect to $S$, that is,

$$2\|(x)_S\|_1 \leq \|x\|_1 \text{ for } x \in \text{Ker}(A) \backslash \{0\}. \tag{3}$$

In the case $A$ has i.i.d. sub-Gaussian entries and $N \gtrsim |S|c^{-2} \log(d/\delta)$ for $c \in (0, 1/2)$, we then have $\|(x)_S\|_1 \leq c\|x\|_1$ with probability greater than $1 - \delta$ [17].

The proof for Theorem 1 proceeds in two steps. Firstly, the TVBP problem (1) is reformulated into a standard basis pursuit problem (2) with an augmented matrix $A$, signal $x^\star$ and support set $S$. Secondly, we show that the Restricted Null Space Property can be satisfied in this case by utilising the structure of $A$, $x^\star$ and $S$. Each of these steps is described within the following paragraphs. For clarity, we assume the TVBP problem with $\widetilde{G} = G$, the signal graph $G$ being a tree graph with agent 1 as the root, and all non root agents with equal design matrix $A_v = \widehat{A}$ for $v \in \{2, 3, 4, 5, \ldots, n\}$. Discussion on weakening these assumptions is provided at the end, with fulls proofs given in Appendix D.

**Reformulating TVBP problem** Let us denote $x_1 \in \mathbb{R}^d$ and $\Delta_i \in \mathbb{R}^d$ for $i = 1, \ldots, n-1$ where edges $e \in E$ are associated to an integer $e \to i$. The TVBP problem (1) can then be reformulated as

$$\min_{x_1, \{\Delta_e\}_{e \in E}} \|x_1\|_1 + \sum_{i=1}^{n-1} \|\Delta_i\|_1 \quad \text{subject to} \quad \begin{pmatrix} A_1 & 0 & 0 & \ldots & 0 \\ \widetilde{A} & \widetilde{A} & H_{13} & \ldots & H_{1n} \\ \widetilde{A} & H_{22} & \widetilde{A} & \ldots & H_{2n} \\ \vdots & \ddots & \ddots & \ddots & \vdots \\ \widetilde{A} & H_{n2} & H_{n3} & \ldots & \widetilde{A} \end{pmatrix} \begin{pmatrix} x_1 \\ \Delta_1 \\ \Delta_2 \\ \vdots \\ \Delta_{n-1} \end{pmatrix} = \begin{pmatrix} y_1 \\ y_2 \\ y_3 \\ \vdots \\ y_n \end{pmatrix}$$

where the matrices $\{H_{ij}\}_{i,j=1,\ldots,n-1}$ take values $H_{ij} = \widetilde{A}$ if the $j$th agent is on the path from agent $i$ to the root node 1, and 0 otherwise. The above is equivalent to a Basis Pursuit problem (2) with

$x = (x_1, \Delta_1, \Delta_2, \ldots, \Delta_n)$, $y = (y_1, y_2, \ldots, y_n)$, an $A \in \mathbb{R}^{nd \times nd}$ encoding the linear constraint above, and sparsity structure $S = S_1 \cup \{\cup_{e,i}(S_e + i)\}$. That is for an edge $e = (v, w) \in E$ the support $S$ is constructed by off-setting $S_e = \text{Supp}(x_v^\star - x_w^\star)$ by an integer $(S_e + i) = \{k + i : k \in S_e\}$.

**Showing Restricted Null Space Property** We begin by noting that if $x \in \text{Ker}(A)\backslash\{0\}$ then

$$A_1 x_1 = 0 \qquad \widetilde{A}\Delta_e = 0 \text{ for } e = (v, w) \in E \text{ such that } v, w \neq 1, \qquad (4)$$

the second equality being over edges $e$ not connected to the root node. To see this, suppose the edge $e \in E$ is both: associated to the integer $i$; and not directly connected to the root $1 \notin e$. Consider the edge neighbouring $e$ closest to the root, say, $e' \in E$ with integer $j$. We have from $x \in \text{Ker}(A)\backslash\{0\}$

$$\widetilde{A}x_1 + \widetilde{A}\Delta_i + \sum_{k \neq i} H_{ik}\Delta_k = 0 \qquad \text{and} \qquad \widetilde{A}x_1 + \widetilde{A}\Delta_j + \sum_{k \neq j} H_{jk}\Delta_k = 0.$$

Taking the difference of the two equations we get $\widetilde{A}\Delta_i = 0$ since both: $j$ is on the path from $i$ to the root so $H_{ij} = \widetilde{A}$; and the path from $j$ to the root is shared i.e. $\sum_{k \neq i,j} H_{ij}\Delta_k = \sum_{k \neq j} H_{jk}\Delta_k$.

In a similar manner to Basis Pursuit, the constraints (4) are used to control the norms $\|(x_1)_{S_1}\|_1$ and $\|(\Delta_e)_{S_e}\|_1$ for $e \in E\backslash\{e \in E : 1 \in e\}$. Precisely, if $A_1 \in \mathbb{R}^{N_1 \times d}$ and $\widetilde{A} \in \mathbb{R}^{N_{\text{Non-root}} \times d}$ both i.i.d. sub-Gaussian with $N_1 \gtrsim s\log(1/\delta)$ and $N_{\text{Non-root}} \gtrsim s'\log(1/\delta)$, then $\|(x_1)_{S_1}\|_1 \leq \|x_1\|_1/4$ and $\|(\Delta_e)_{S_e}\|_1 \leq \|\Delta_e\|_1/4$ with high probability. Controlling the norm for the edges $e \in E$ connected to the root $1 \in e$ is then more technical. In short, if $N_1, N_{\text{Non-Root}} \gtrsim \text{Deg}(1)^2 s'\log(1/\delta)$ then $\|(\Delta_e)_{S_e}\|_1 \leq (\|x_1\|_1 + \|\Delta_e\|_1)/4\text{Deg}(1)$. Summing up the bounds gives

$$\|(x)_S\|_1 = \|(x_1)_{S_1}\|_1 + \sum_{e \in E} \|(\Delta_e)_{S_e}\|_1 \leq \frac{1}{2}\|x_1\|_1 + \frac{1}{2}\sum_{e \in E}\|\Delta_e\|_1 = \frac{1}{2}\|x\|_1$$

as required. When the matrices $\{A_v\}_{v \in V\backslash\{1\}}$ are different the condition (4) may no longer be satisfied, and thus, an alternative analysis is required. Meanwhile, if $\widetilde{G}$ is a star and does not equal $G$, a different sparsity set $\widetilde{S}$ is considered where the support along edges are swapped $s' \to \text{Diam}(G)s'$.

# 3 Noisy Setting

This section demonstrates how the TVBP problem can be extended to the noisy setting, Section 3.1 introduces Total Variation Basis Pursuit Denoising (TVBPD) alongside theoretical guarantees. Section 3.2 presents experiments investigating the performance of TVBPD.

## 3.1 Total Variation Basis Pursuit Denoising

Let us assume that $y_v = A_v x_v^\star + \epsilon_v$ for $v \in V$. In this case the equality constraint in the TVBP problem (1) is swapped for a softer penalisation, leading to the *Total Variation Basis Pursuit Denoising* (TVBPD) problem for a graph $\widetilde{G} = (V, \widetilde{E})$ and penalisation $\eta > 0$

$$\min_{\{x_v\}_{v \in V}} \|x_1\|_1 + \sum_{e=(v,w)\in\widetilde{E}} \|x_v - x_w\|_1 \text{ subject to } \sum_{v \in V} \|A_v x_v - y_v\|_2^2 \leq \eta^2. \qquad (5)$$

The equality constraint $A_v x_v = y_v$ at each agent $v \in V$ in (1) is now swapped with an upper bound on the deviation $\|A_v x_v - y_v\|_2^2$. Given this problem, we now provide gaurantees on the $\ell_1$ estimator error for the solution of (5). The proof of the following Theorem is in Appendix 3.1.

**Theorem 2.** *Suppose $G$ is a tree graph, the signals $\{x_v^\star\}_{v \in V}$ are $(G, s, s')-$sparse and $y_v = A_v x_v^\star + \epsilon_v$ for $v \in V$. Assume that $A_v = \widetilde{A}_v/\sqrt{N_v}$ where $\{\widetilde{A}_v\}_{v \in V}$ each have i.i.d. sub-Gaussian entries. Fix $\epsilon > 0$. If $\eta^2 \geq \sum_{v \in V}\|\epsilon_v\|_2^2/n$ and*

$$N_{Root} \gtrsim s\big(\log(d) + \log(1/\epsilon)\big) \quad and \quad N_{Non-root} \gtrsim n^2 s'\big(\log(d) + \log(n/\epsilon)\big),$$

*then with probability greater than $1 - \epsilon$ the solution to (5) with $\widetilde{G} = G$ satisfies*

$$\|x_1 - x_1^\star\|_1 + \sum_{e=(v,w)\in E} \|(x_v - x_w) - (x_v^\star - x_w^\star)\|_1 \lesssim \big(\sqrt{s} + Deg(G)\sqrt{ns'}\big)\eta.$$

Theorem 2 gives conditions on the sample size so a bound on the $\ell_1$ estimation error can be achieved. For the stepwise approach in Section 2.1 the $\ell_1$ estimation error scales as $\sqrt{s} + n \times \sqrt{s'}$. Therefore, TVBPD yields an order $\sqrt{n}$ saving in $\ell_1$ estimation error over the step wise approach. This highlights two sample size regimes. When the total sample size is $O(s + ns')$, the step wise approach is provably feasible and the estimation error is $O(\sqrt{s} + n\sqrt{s'})$. Meanwhile, when the total sample size is $O(s + n^3 s')$, TVBPD is provably feasible and the estimation is $O(\sqrt{s} + \sqrt{ns'})$. The gap in sample size requirements between the noisy case with TVBPD $O(s + n^3 s')$ and the noiseless case with TVBP $O(s + n^2 \text{Diam}(G) s')$ is due to a different proof technique for TVBPD. We leave extending the techniques for analysing TVBP to the case of TVBPD to future work.

### 3.2  Experiments for Total Variation Basis Pursuit Denoising

This section present simulation results for the Total Variation Basis Pursuit Denoising problem (5). The following paragraphs, respectively, describe results for synthetic and real data.

**Synthetic Data.** Figure 3 plots the $\ell_1$ estimation error for Total Variation Basis Pursuit Denoising, group Lasso and the Dirty Model of [24], against the number of agents for both path and balanced tree topologies. As the number of agents grows, the estimation error for the group Lasso methods grows quicker than the total variation approach. The group Lasso variants perform poorly here due to the union of supports growing with the number of agents, and thus, the small overlap between agent's supports.

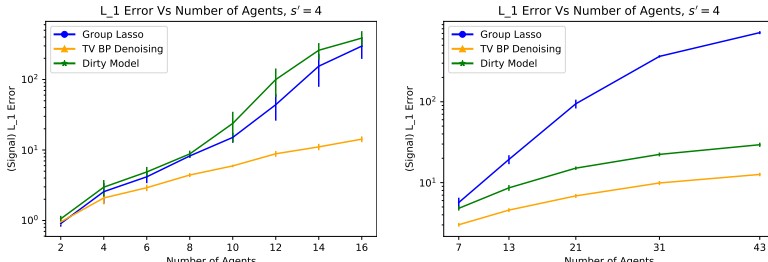

Figure 3: $\ell_1$ estimation error $\sum_{v \in V} \|x_v - x_v^\star\|_1$ ($\log_{10}$ scale) against number of agents for Total Variation Basis Pursuit Denoising solved using SPGL1 Python package (*Yellow*), group Lasso (*blue*) and Dirty Model of [24] (*Green*). *Left*: Path topology. *Right*: Balanced tree topology height 2 branching rate $\{2, 3, 4, 5, 6\}$. The same i.i.d. standard Gaussian matrix was associated to each node with $N_v = 200$ for $v \in V$, with parameters were $d = 2^9$, $s = 25$ and $s' = 4$. Signal at the root $x_1^\star$ and differences $\{x_v^\star - x\star_w\}_{(v,w) \in E}$ sampled from $\{+1, -1\}$ with no overlap in supports.

**Hyperspectral Unmixing.** We apply Total Variation Basis Pursuit Denoising to the popular AVIRIS Cuprite mine reflectance dataset `https://aviris.jpl.nasa.gov/data/free_data.html` with a subset of the USGS library splib07 [26]. As signals can be associated to pixels in a 2-dimensional image, it is natural to consider the total variation associated with a grid topology. Computing the total variation explicitly in this case can be computationally expensive, see for instance [39]. We therefore simplify the objective by tilling the image into groups of $n = 4$ pixels arranged in a 2x2 grid, with each group considered independently. This is common approach within parallel rendering techniques, see for instance [34], and is justified in our case as the signals are likely most strongly correlated with their neighbours in the graph. Note that this also allows our approach to scale to larger images as the algorithm can be run on each tile in an embarrassingly parallel manner. More details of the experiment are in Appendix B.2.

We considered four methods: applying Basis Pursuit Denoising to each pixel independently; Total Variation Denoising (5) applied to the groups of 4 pixels as described previously; the group Lasso applied to the groups of $4$ pixels described previously.; and a baseline Hyperspectral algorithm SUNnSAL [2]. Figure 4 then gives plots of the coefficients associated to two minerals for three of the methods. Additional plots associated to four minerals and the four methods have been Figure 6 Appendix B.2. Recall, by combining pixels the aim is to estimate more accurate coefficients than from denoising them independently. Indeed for the Hematite, Andradite and Polyhalite minerals, less noise is present for the total variation approach, alongside larger and brighter clusters. This is also in comparison to SUNnSAL, where the images for Andradite and Polyhalite from the total variation

approach have less noise and brighter clusters. Although, we note that combining groups of pixels in this manner can cause the images to appear at a lower resolution.

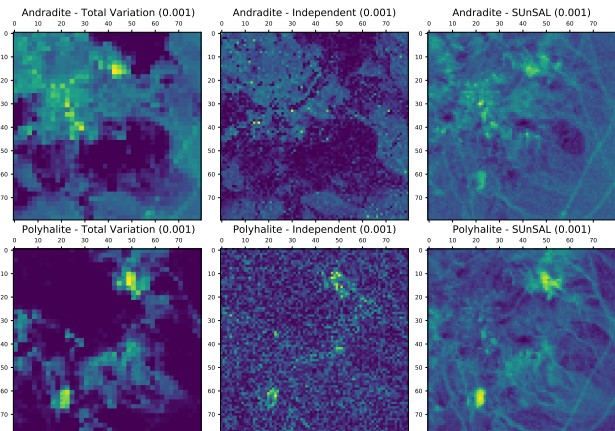

Figure 4: Coefficients associated to the minerals Andradite (*top*), and Polyhalite (*bottom*). Methods are, *left*: Total Variation Basis Pursuit Denoising applied to 2x2 pixel tiles with $\eta = 0.001$; *Middle*: Basis Pursuit Denoising applied independently to each pixel with $\eta = 0.001$. *right*: SUNnSAL with regularisation of $0.001$. Yellow pixels indicate higher values.

## 4   Conclusion

In this work we investigated total variation penalty methods to jointly learn a collection of sparse linear models over heterogeneous data. We assumed a graph-based sparse structure for the signals, where the signal at the root and the signal differences along edges are sparse. This setting differs from previous work on solving collections of sparse problems, which assume large overlapping supports between signals. We demonstrated (in noiseless and noisy settings) that statistical savings can be achieved over group Lasso methods as well as solving each problem independently, in addition to developing a distributed ADMM algorithm for solving the objective function in the noiseless case.

The theoretical results currently suggest having identical matrices for non-root agents is more sample efficient over having different matrices (Table 2 and proof sketch Section 2.4). A natural direction is to investigate whether this is a limitation of the analysis, or fundamental to using the Total Variation penalty. Following this work, a distributed ADMM algorithm can also be developed for Total Variation Basis Pursuit Denoising optimisation problem (5).

## Acknowledgements

D.R. is supported by the EPSRC and MRC through the OxWaSP CDT programme (EP/L016710/1), and the London Mathematical Society ECF-1920-61. S.N. was supported by NSF DMS 1723128. P.R. was supported in part by the Alan Turing Institute under the EPSRC grant EP/N510129/1.

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
