# A   Additional Material - Noiseless setting

In this section we present additional material associated to the noiseless setting. Section A.1 presents refined sample complexity results when additional assumptions are placed on the graph $\widetilde{G}$ or the design matrices $\{A_v\}_{v \in V \setminus \{1\}}$. Section A.2 presents details related to the distributed ADMM algorithm presented within Section 2.3 of the main body.

## A.1   Refined Theoretical Results for Total Variation Basis Pursuit

In this section we present refined results for Total Variation Basis Pursuit. Section A.1.1 consider the case of a tree graph with different design matrices at each node. Section A.1.2 considers the case of a tree graph with identical design matrices at non-root agents.

### A.1.1   Total Variation Basis Pursuit with known Tree Graph

Let us now consider the case where $G$ is a known tree graph. The sample complexity in this case is summarised within the follow Theorem.

**Theorem 3.** *Suppose $G$ is a tree graph, the signals $\{x_v^\star\}_{v \in V}$ are $(G, s, s')$-sparse and matrices satisfy $A_v = \frac{1}{\sqrt{N_v}} \widetilde{A}_v$ where $\{\widetilde{A}_v\}_{v \in V}$ have i.i.d. sub-Gaussian entries. Fix $\epsilon > 0$. If*

$$N_{Root} \gtrsim \max\{s, n^2 s'\}\big(\log(d) + \log(1/\epsilon)\big) \text{ and}$$
$$N_{Non\text{-}root} \gtrsim \max\big\{n, Deg(V \setminus \{1\})^2 Diam(G)^2\big\} s'\big(\log(d) + \log(n/\epsilon)\big)$$

*then with probability greater than $1 - \epsilon$ the solution $\{x_v^{TVBP}\}_{v \in V}$ with $G = \widetilde{G}$ is unique and satisfies $x_v^{TVBP} = x_v^\star$ for all $v \in V$.*

*Proof.* See Appendix D.5. $\qquad\qquad\qquad\qquad\qquad\qquad\qquad\qquad\qquad\qquad\qquad\qquad\square$

The sample complexity listed within the second row of Table 2 is then arrived at by simply summing up the above bound to arrive at $N_{\text{Total Samples}} = O(s + \max\{n^2 s', n Deg(V \setminus \{1\})^2 Diam(G)^2\} s')$. We then note that Theorem 1 within the main body of the work is a direct consequence of Theorem 3. We formally presented these steps within the following proof.

*Proof of Theorem 1.* Let us consider a signal $\{x_v^\star\}_{v \in V}$ that is $(G, s, s')-$sparse with respect to a general graph $G$. We then see that the signal is then $(\widetilde{G}, s, Diam(G)s')$-sparse when $\widetilde{G}$ is a star topology. Using Theorem 3 there after (swapping $G$ for $\widetilde{G}$ and $s'$ for $Diam(G)s'$ and noting that both $Diam(\widetilde{G}) = 1$ and the degree of the non-root agents is $Deg(V \setminus \{1\}) = 1$ yields the result. $\quad\square$

### A.1.2   Tree Graph with Identical Matrices for Non-root Agents

Let us now consider when non-root agents have the *same* sensing matrices i.e $A_v = A_w$ for $v, w \in V \setminus \{1\}$. The result is then summarised within the follow Theorem 4.

**Theorem 4.** *Suppose $G$ is a tree graph, $\{x_v^\star\}_{v \in V}$ is $(G, s, s')$-sparse, $A_v = \frac{1}{\sqrt{N_{Non\text{-}root}}} A_{Non\text{-}root}$ for $v \neq 1$ and $A_1 = \frac{1}{\sqrt{N_1}} A_{Root}$. Assume that $A_{Root}, A_{Non\text{-}root}$ each have i.i.d. sub-Gaussian entries. Fix $\epsilon > 0$. If*

$$N_{Root} \gtrsim \max\{s, Deg(1)^2 s'\}\big(\log(d) + \log(1/\epsilon)\big) \text{ and}$$
$$N_{Non\text{-}root} \gtrsim Deg(1)^2 s'\big(\log(d) + \log(1/\epsilon)\big)$$

*then with probability greater than $1 - \epsilon$ the solution to TVBP with $\widetilde{G} = G$ is unique and satisfies $x_v^{TVBP} = x_v^\star$ for all $v \in V$.*

*Proof.* See Appendix D.4. $\qquad\qquad\qquad\qquad\qquad\qquad\qquad\qquad\qquad\qquad\qquad\qquad\square$

The entry within the third row of Table 2 is then arrived at by summing up the above bound to arive at $N_{\text{Total Samples}} = O(s + n Deg(1)^2 s')$.

## A.2 Distributed ADMM Algorithm

In this section present the Distributed ADMM algorithm for solving the Total Variation Basis Pursuit problem (1). We begin by reformulating the problem into an consensus optimisation form. Specifically, with $\Delta_e = x_v - x_w$ for $e = \{v, w\} \in E$, we consider

$$\min_{x_v, v \in V} \|x_1\|_1 + \sum_{e \in V} \|\Delta_e\|_1 \text{ subject to}$$

$$A_v x_v = Y_v \text{ for all } v \in V \text{ and } x_v - x_w = \Delta_e \text{ for all } e = \{v, w\} \in E.$$

We then propose the Alternating Direction Method of Multipliers (ADMM) to solve the above. The key step is consider the augmented Lagrangian from dualizing the consensus constraint which, with $\|x\|_1 = \|x_1\|_1 + \sum_{e \in E} \|\Delta_e\|_1$, is for $\rho > 0$

$$\mathcal{L}_\rho(\{x_v\}_{v \in V}, \{\Delta_e\}_{e \in E}, \{\gamma_e\}_{e \in E}) = \|x\|_1 + \sum_{e = \{v, w\} \in E} \frac{\rho}{2} \|x_v - x_w - \Delta_e\|_2 + \langle \gamma_e, x_v - x_w - \Delta_e \rangle.$$

The ADMM algorithm then proceeds to minimise $\mathcal{L}_\rho$ with respect to $\{x_v\}_{v \in V}$, then $\{\Delta_e\}_{e \in E}$, followed by a ascent step in the dual variable $\{\gamma_e\}_{e \in E}$. Full details of the ADMM updates have been given in Appendix C. Each step can be computed in closed form, expect for the update for $x_1$ which requires solving a basis pursuit problem with an $\ell_2$ term in the objective. This can be solved to a high precision efficiently by utilising a simple dual method, see [35, Appendix B]. The additional computational required by the root node in this case aligns with the framework we consider, since we assume the root node also has an additional number of samples $N_1$.

The theoretical convergence guarantees of ADMM have gained much attention lately due to the wide applicability of ADMM to distributed optimisation problems [4, 18, 20]. While a full investigation of the convergence guarantees of ADMM in this instance is outside the scope of this work, we note for convex objectives with proximal gradient steps computed exactly, ADMM has been shown to converge at worst case a polynomial rate of order $1/t$ [18]. A number of works have shown linear convergence under additional assumptions which include full column rank on the constraints or strong convexity, which are not satisfied in our case [1]. Although, if one considers a proximal variant of ADMM with an additional smoothing term, linear convergence can be shown in the absence of the column rank constraint [20]. The convergence of ADMM can be sensitive hyperparameter choice $\rho$, which has motivated a number of adaptive schemes, see for instance [19].

# B Additional Material - Noisy Setting

In this section we present additional material associated to the noisy setting within the main body of the manuscript. Section B.1 presents details for experiments on simulated data. Section B.2 details for the experiments on real data.

## B.1 Details for Total Variation Basis Pursuit Denoising - Simulated Data

We now provide some details related to the **Simulated Data** experiments in Section 3.1 the manuscript. Group Lasso used best regularisation from between $[10^{-6}, 10^{-2}]$. Dirty model regularisation followed [24] with (in their notation) $5 \times 5$ (log -scale) grid search for $\lambda_g$ and $\lambda_b$ with $\lambda_g/\lambda_b \in [10^{-3}, 10]$, $\lambda_b = c\sqrt{7/200}$ and $c \in [10^{-2}, 10]$. Dirty model was fit using MALSAR [56]. The group Lasso variants used normalised matrices $A_v/\sqrt{N_v}$ and responses $y_v/\sqrt{N_v}$. Total Variation Basis Pursuit Denoising parameter was $\eta = \sqrt{200 \times n}0.1$. Each point and error bars from 5 replications.

## B.2 Data Preparation and Experiment Parameters for AVIRIS Application

In this section we present details associated to the application of Total Variation Basis Pursuit Denoising TVBPD (5) to the AVIRIS Cuprite dataset. We begin with Figure 5, which presents the sector of the AVIRIS Cuprite dataset used, as well as the 80 x 80 pixel subset portion sub-sampled

---

[1] The constraint dualised by ADMM, $x_v - x_w = \Delta_e$ for $e = \{v, w\} \in E$, can be denoted in terms the signed incident matrix of the graph. This is a linear constraint, but the signed incident matrix does not have full column rank.

for our experiment. We note each pixel in the dataset is associated to 224 spectral bands between 400 and 2500 nm and, in short, the objective is to decompose the spectrum of each pixel into a sparse linear combination known mineral spectra. The specific bandwidth presented in Figure 5 demonstrate that this area maybe a region of interest. Following [22, 23], we construct a spectral library $A_{\mathrm{Lib}}$ by randomly sampling 240 mineral from the USGS library splib07 [2]. After cleaning the AVIRIS dataset and the library we are left with $N_v = 184$ spectral bands for each pixel $v \in V$, and thus, $A_v = A_{\mathrm{Lib}} \in \mathbb{R}^{184 \times 240}$ and $y_v \in \mathbb{R}^{184}$. We now go on to describe more detail the experimental steps.

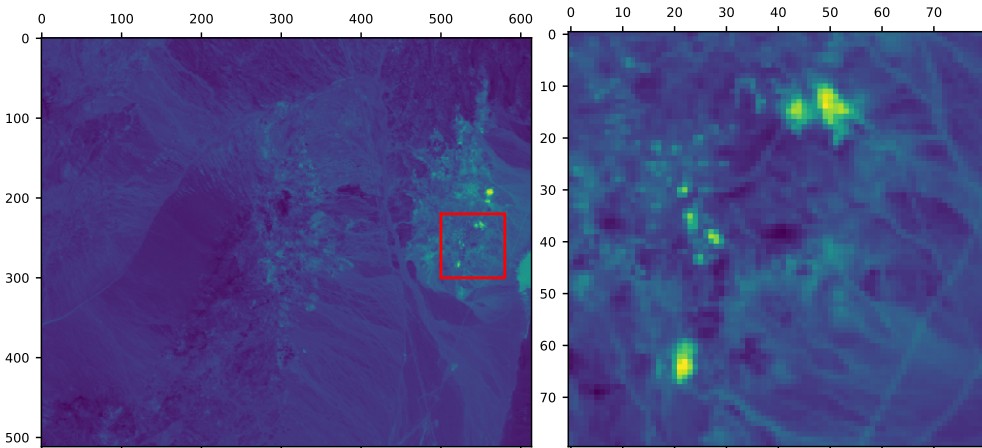

Figure 5: *Left*: Sector `f970619t01p02_r04_sc03.a.rfl` of AVIRIS data set at bandwidth of 557.07 nm. Red square indicates $80 \times 80$ portion of the sector used as the data set. *Right*: Red squared section zoomed in.

**Cleaning AVIRIS Cuprite Dataset** We followed [23] and removed the spectral bands 1-2, 105-115, 150-170 and 223-224, which are due to water absorption and low signal to noise. This would leave us with 188 spectral bands, although additional bands were removed due to large values within the USGS Library, see next paragraph.

**Sub-sampling USGS Library** We took a random sample of 240 minerals from splib07 library, that are specifically calibrated to the AVIRIS 1997 data set i.e. have been resampled at the appropriate bandwidths. A number of the spectrum for the minerals were corrupted or had large reflectance values for particular wavelengths e.g. greater than $10^{34}$. We therefore restricted ourselves to minerals that had less than 10 corrupted wavelengths. After sub-sampling, any wavelengths with a corrupted value (if it contained a value greater than 10) were removed. This left us with 184 spectral bands.

**Algorithm Parameters** To apply Basis Pursuit Denoising independently to each pixel, we used the SPGL1 python package, which can be found at `https://pypi.org/project/spgl1/`. To solve the Total Variation Basis Pursuit Denoising problem (5), we used the Alternating Direction Methods of Multiplers (ADMM) algorithm for $\ell_1$-problems in [54], specifically the inexact method (2.16). We applied this algorithm to the normalised data i.e. dividing by the matrix and response vector by the square root of the total number of samples (4 pixels $\times$ 184 spectral bands). We ran the algorithm for 500 iterations with parameters (in the notation of [54]) $\tau = 0.1$, $\beta = 2$, $\gamma = 0.1$ and $\delta = 0.001$. We note that directly applying the SPGL1 python package to the Total Variation Basis Pursuit Denoising problem (5), resulted in instabilities when choosing $\eta < 0.2$. We chose $\eta = 0.001$ for both independent Basis Pursuit Denoising case and the Total Variation Basis Pursuit Denoising (5), following the regularisation choice in [23]. Meanwhile, the group Lasso was fit using scikit-learn with regularisation 0.001, and the SUNnSAL algorithm [2] with regularisation 0.001 was applied using the python implementation which can be found at `https://github.com/Laadr/SUNSAL`. We note when using SUNnSAL it is common to perform a computationally expensive pre-processing

---

[2]`https://crustal.usgs.gov/speclab/QueryAll07a.php`

step involving a non-convex objective, see [22, 23]. This was not performed in this case, as all of the other methods did not pre-process the data.

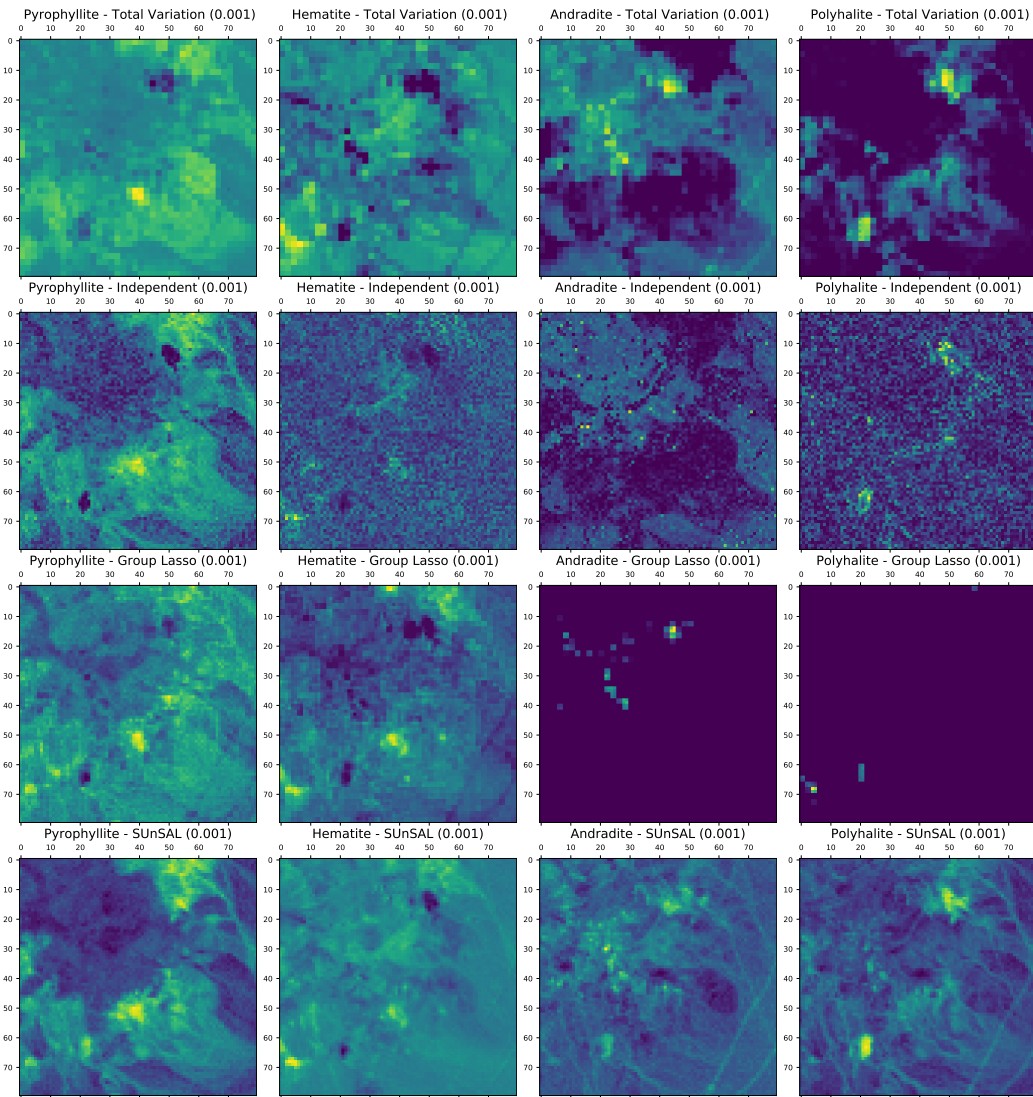

Figure 6: Coefficients associated to the mineral Pyrophyllite (*Left*), Hematite (*Left-Middle*), Andradite (*right-middle*), and Polyhalite (*Right*). Methods considered are: *Top*: Total Variation Basis Pursuit Denoising applied to 2x2 pixels simultaneously with $\eta = 0.001$; *Middle-Top*: Basis Pursuit Denoising applied independently to each pixel with $\eta = 0.001$. *Middle-Bottom*: group Lasso (jointly penalised all coefficients) applied to 2x2 pixels simultaneously with regularisation 0.001. *Bottom*: SUnSAL with regularisation of 0.001. Yellow pixels indicate higher values.

## C  Distributed ADMM Updates for Total Variation Basis Pursuit

In this section we more precisely describe the Distributed ADMM algorithm for fitting the Total Variation Basis Pursuit problem (11). We recall the consensus optimisation formulation of the Total Variation Basis Pursuit problem is as follows

$$\min_{x_v, v \in V} \|x_1\|_1 + \sum_{e \in V} \|\Delta_e\|_1 \text{ subject to}$$

$$A_v x_v = Y_v \text{ for all } v \in V$$

$$x_v - x_w = \Delta_e \text{ for all } e = \{v, w\} \in E.$$

where we consider the Augmented Lagrangian from dualizing the consensus constraint

$$\mathcal{L}_\rho(\{x_v\}_{v\in V}, \{\Delta_e\}_{e\in E}, \{\gamma_e\}_{e\in E}) = \|x_1\|_1$$
$$+ \sum_{e=\{v,w\}\in E} \|\Delta_e\|_1 + \frac{\rho}{2}\|x_v - x_w - \Delta_e\|_2 + \langle\gamma_e, x_v - x_w - \Delta_e\rangle.$$

Now the ADMM algorithm initialized at $\left(\{x_v^1\}_{v\in V}, \{\Delta_e^1\}_{e\in E}, \{\gamma_e^1\}_{e\in E}\right)$ then proceeds to update the iterates for $t \geq 1$ as

$$x_v^{t+1} = \underset{x_v^t}{\arg\min}\, \mathcal{L}_\rho(\{x_v^t\}_{v\in V}, \{\Delta_e^t\}_{e\in E}, \{\gamma_e^t\}_{e\in E}) \text{ subject to } A_v x_v = Y_v \text{ for } v\in V \qquad (6)$$

$$\Delta_e^{t+1} = \underset{\Delta_v^t}{\arg\min}\, \mathcal{L}_\rho(\{x_v^{t+1}\}_{v\in V}, \{\Delta_e^t\}_{e\in E}, \{\gamma_e^t\}_{e\in E}) \qquad\qquad\qquad \text{for } e\in E$$

$$\gamma_e^{t+1} = \gamma_e^t + \rho\big(x_v - x_w - \Delta_e\big) \qquad\qquad\qquad\qquad\qquad\qquad \text{for } e\in E$$

We now set to show how each of the above updates can be implemented in a manner that respects the network topology due to the Augmented Lagrangian $\mathcal{L}_\rho$ decoupling across the network. These will be precisely described within the following sections. For clarity each update will be given its own subsection and the super script notation i.e. $x_v^t$ will be suppressed.

## C.1 Updating $\{x_v\}$

The updates for $\{x_v\}_{v\in V}$ take two different forms depending on whether $v$ is associated to the root node i.e. $v = 1$ or otherwise. We begin with the case of a root note.

### C.1.1 Root Node $x_1$

The update for $x_1$ in the ADMM algorithm (6) requires solving

$$\min_{x_1} \|x_1\|_1 + \sum_{e=(i,j)\in E: i=1} \frac{\rho}{2}\|x_1 - x_j - \Delta_e\|_2^2 + \langle\gamma_e, x_1 - x_j - \Delta_e\rangle$$
$$+ \sum_{e=(i,j)\in E: j=1} \frac{\rho}{2}\|x_j - x_1 - \Delta_e\|_2^2 + \langle\gamma_e, x_j - x_1 - \Delta_e\rangle$$

$$\text{subject to } A_1 x_1 = y_1$$

where we note the two summations in the objective arise from the orientation of the edges within the network. This is then equivalent to considering solve a problem of the form

$$\min_x \|x\|_1 + \nu^\top x + c\|x\|_2^2 \text{ subject to } Ax = b \qquad (7)$$

with parameters $A = A_1$, $b = y_1$, $c = \text{Deg}(1)\frac{\rho}{2}$ where $\text{Deg}(1)$ is the degree of the root node 1 and $\nu = \sum_{e=(i,j)\in E: i=1} \gamma_e - \rho(x_i + \Delta_e) + \sum_{e=(i,j)\in E: j=1} -\gamma_e - \rho(x_j + \Delta_e)$.

To solve the problem (7) we adopt the approach used in [35, Appendix B] to an optimisation problem of the same form. That is, we consider the dual problem

$$\max_\lambda \lambda^\top b + \sum_{i=1}^p \inf_{x_i} \left(|x_i| + u_i(\lambda)x_i + cx_i^2\right)$$

where the dual variable $\lambda \in \mathbb{R}^n$ and $u(\lambda) = \nu - A^\top\lambda$. The gradient of the above problem is then $b - Ax(\lambda)$ where $x(\lambda) = (x(\lambda)_1, \ldots, x(\lambda)_p)$ is constructed from the unique minimiser of $|x_i| + u_i(\lambda)x_i + cx_i^2$ for $i = 1, \ldots, p$ which is $x(\lambda)_i$. This can then be written in closed form as

$$x_i(\lambda) = \begin{cases} 0 & \text{if } -1 \leq u_i(\lambda) \leq 1 \\ -(u_i(\lambda) + 1)/2c & \text{if } u_i(\lambda) < -1 \\ -(u_i(\lambda) - 1)/2c & \text{if } u_i(\lambda) > 1 \end{cases}$$

Given a solution $\lambda^\star$ the solution to the original problem is then $x(\lambda^\star)$. To solve the Dual problem we use the Barzilai - Borwein algorithm [41] with warm restarts using the dual variable from the previous iteration.

### C.1.2 Non-Root Node

In the case of $x_v$ which is not the root node i.e. $v \neq 1$, we require solving the optimisation problem

$$\min_{x_v} \sum_{e=(i,j)\in E:i=v} \frac{\rho}{2}\|x_v - x_j - \Delta_e\|_2^2 + \langle \gamma_e, x_v - x_j - \Delta_e \rangle$$

$$+ \sum_{e=(i,j)\in E:j=v} \frac{\rho}{2}\|x_i - x_v - \Delta_e\|_2^2 + \langle \gamma_e, x_i - x_v - \Delta_e \rangle$$

subject to $\quad A_v x_v = y_v$

This minimisation can be written in the form

$$\min_x \|x\|_2^2 + \langle a, x \rangle \text{ subject to} \qquad (8)$$

$$Ax = b$$

with parameters $A = A_v$, $b = y_v$ and
$a = \frac{2}{\text{Deg}(v)}\Big(\big(\sum_{e\in\{i,j\}:i=v} -\Delta_e - x_j + \frac{\gamma_e}{\rho}\big) + \big(\sum_{e\in\{i,j\}:j=v}\Delta_e - x_i - \frac{\gamma_e}{\rho}\big)\Big)$. Since $\|x\|_2^2 + \langle a, x \rangle = \|x + \frac{a}{2}\|_2^2 - \frac{1}{2}\|a\|_2^2$, This leads to the equivalent optimisation problem

$$\min_u \|u\|_2^2 \text{ subject to}$$

$$Au = b + A\frac{a}{2}.$$

This is exactly the least norm solution to a linear system, and is solved by $u = A^\dagger(b + A\frac{a}{2})$ where $A^\dagger$ is the Moore-Penrose pseudo-inverse. We then recover the solution to (8) by setting $x = A^\dagger(b + A\frac{a}{2}) - \frac{a}{2}$.

### C.2 Updating $\{\Delta_e\}_{e\in V}$

For each edge $e = (i, j) \in E$ the updates require solving

$$\min_{\Delta_e} \|\Delta_e\|_1 + \frac{\rho}{2}\|x_i - x_j - \Delta_e\|_2^2 - \langle \gamma_e, \Delta_e \rangle$$

which is a equivalent to

$$\min_{\Delta_e} \|\Delta_e\|_1 + \frac{\rho}{2}\|\Delta_e\|_2^2 - \langle \Delta_e, \gamma_e + z_i - z_j \rangle.$$

This is a shrinkage step and thus the minimiser can be written as

$$\Delta_e = \begin{cases} 0 & \text{if } |\gamma_e + \rho(z_i - z_j)| < 1 \\ \frac{1}{\rho}\big(\gamma_e + \rho(z_i - z_j) - 1\big) & \text{if } \gamma_e + \rho(z_i - z_j) > 1 \\ \frac{1}{\rho}\big(\gamma_e + \rho(z_i - z_j) + 1\big) & \text{if } \gamma_e + \rho(z_i - z_j) < -1 \end{cases}$$

## D  Proofs For Noiseless Case

In this section we present proofs for the results associated to Total Variation Basis Pursuit (TVBP) (1). This section is then structured as follows. Section D.1 presents technical lemmas associated to the Restricted Isometry Property of matrices. Section D.2 introduces the Basis Pursuit problem. Section D.3 demonstrates how the Total Variation Basis Pursuit (TVBP) problem can be reformulated into a Basis Pursuit problem. Section D.4 presents the proof of Theorem 4. Section D.5 presents the proof of Theorem 3.

### D.1 Technical Lemmas for the Restricted Isometry Property

Recall that a matrix $A \in \mathbb{R}^{N \times d}$ satisfies Restricted Isometry Property at level $k$ if there exists a constant $\delta_k \in [0, 1)$ such that for any $k$-sparse vector $x \in \mathbb{R}^p$, $\|x\|_0 \leq k$ we have

$$(1 - \delta_k)\|x\|_2^2 \leq \|Ax\|_2^2 \leq (1 + \delta_k)\|x\|_2^2$$

Now, Theorem 9.2 from [17] demonstrates that a sub-Gaussian matrix can satisfy the Restricted Isometry Property in high probability provided the sample size is sufficiently large. This is presented within the following theorem.

**Theorem 5.** *Let $A \in \mathbb{R}^{N \times p}$ be sub-Gaussian matrix with independent and identically distributed entries. Then there exists a constant $C > 0$ (depending on sub-Gaussian parameters $\beta$ and $\kappa$) such that the Restricted Isometry Constant of $A/\sqrt{N}$ satisfies $\delta_k \leq \delta$ with probability atleast $1 - \epsilon$ provided*

$$N \geq C\delta^{-2}\big(k\log(ep/k) + \log(\epsilon/2)\big).$$

We will also make use of the following assumption.

**Proposition 1.** *Let $\mathbf{u}, \mathbf{v} \in \mathbb{R}^p$ be vectors such that $\|\mathbf{u}\|_0 \leq s$ and $\|\mathbf{v}\|_0 \leq k$, and matrix $A \in \mathbb{R}^{N \times p}$ satisfy Restricted Isometry Property up to $s + k$ with constant $\delta_{s+k}$. If the support of the vectors is disjoint $Supp(v) \cap Supp(u) = \emptyset$ then*

$$|\langle A\mathbf{u}, A\mathbf{v}\rangle| \leq \delta_{s+k}\|\mathbf{u}\|\|\mathbf{v}\|_2$$

We also make use of the following lemma which is can be found as Lemma 11 in the Supplementary material of [30]. It will be useful to denote the $\ell_1$ ball as $\mathbb{B}_1(r) = \{x \in \mathbb{R}^d : \|x\|_1 \leq r\}$ and similarly for $\ell_2$ and $\ell_0$ balls as $\mathbb{B}_2(r), \mathbb{B}_0(r)$, respectively.

**Lemma 1.** *For any integer $s \geq 1$, we have*

$$\mathbb{B}_1(\sqrt{s}) \cap \mathbb{B}_2(1) \subseteq \mathrm{cl}\Big\{\mathrm{conv}\Big\{\mathbb{B}_0(s) \cap \mathbb{B}_2(3)\Big\}\Big\}$$

*where* cl *and* conv *denote the topological closure and convex hull, respectively.*

## D.2 Basis Pursuit

For completeness we recall some fundamental properties of the Basis Pursuit problem. Consider a sparse signal $x^\star$, sensing matrix $A \in \mathbb{R}^{N \times d}$ and response $y = Ax^\star$. The Basis Pursuit problem is then defined as

$$\|x\|_1 \text{ subject to} \tag{9}$$
$$Ax = y.$$

Denote the solution to the above as $x^{\mathrm{BP}}$. Suppose that $x^\star$ is supported on the set $S \subset \{1, \ldots, d\}$. Then it is well know [17] that $x^{\mathrm{BP}}$ is both unique and satisfies $x^{\mathrm{BP}} = x^\star$ if and only if $A$ satisfies the *restricted null space property* with respect to $S$, that is,

$$2\|(x)_S\|_1 \leq \|x\|_1 \text{ for any } x \in \mathrm{Ker}(A)\backslash\{0\}. \tag{10}$$

Following the proof sketch in the manuscript, we now proceed to reformulate the Total Variation Pursuit Problem (1) into a Basis Pursuit problem (9).

## D.3 Reformulating Total Variation Basis Pursuit into Basis Pursuit

We now describe the steps in reformulating the Total Variation Basis Pursuit Denoising problem (1) into a Basis Pursuit problem (9). We begin by introducing some notation. For node $v \in V$, denote the set of edges making a path from node $v$ to the root node 1 by $\pi(v) = \{\{v, w_1\}, \{w_1, w_2\}, \ldots, \{w_{k_v-1}, w_{k_v}\}, \{w_{k_v}, 1\}\} \subseteq E$ where $k_v \geq 1$ is the number of intermediate edges. In the case $k_v = 0$ there is only a single edge and so we write $\pi(v) = \{v, 1\} \in E$. Meanwhile, for the root node itself $v = 1$ we simply have the singleton $\pi(v) = \pi(1) = \{1\}$, and thus, we have the root node included $v \in \pi(v)$ but no edges i.e. $e \notin \pi(v)$ for any $e \in E$. For each edge $e = \{v, w\} \in E$ the difference is denoted $\Delta_e = x_v - x_w$, and so the vector associated to any node $x_v$ can be decomposed into the root node $x_1$ plus the differences along the path $x_v = x_1 + \sum_{e \in \pi(v)} \Delta_e$. Similarly, the signal associated to each node $x_v^\star$ can be decomposed into differences of signals associated to the edges $e = \{v, w\} \in E$ with $\Delta_e^\star = x_v^\star - x_w^\star$.

With this notation we can then reformulate (1) in terms of $x_1$ and $\{\Delta_e\}_{e \in E}$ as follows

$$\min_{x_1, \{\Delta_e\}_{e \in E}} \|x_1\|_1 + \sum_{e=(v,w) \in E} \|\Delta_e\|_1 \text{ subject to} \tag{11}$$
$$A_v\Big(x_1 + \sum_{e \in \pi(v)} \Delta_e\Big) = y_v \quad \forall v \in V.$$

Optimisation problem (11) is now in terms of a standard basis pursuit problem (2) with, if edges are labeled with integers, the vector $x = (x_1, \Delta_1, \ldots, \Delta_{|E|})$, true signal $x^\star = (x_1^\star, \Delta_1^\star, \ldots, \Delta_{|E|}^\star)$, and a matrix $A$. To be precise, the matrix $A$ can be defined in terms of blocks $A = (H_1^\top, \ldots, H_n^\top)^\top \in \mathbb{R}^{(\sum_{v \in V} N_v) \times np}$ with each block $H_v \in \mathbb{R}^{N_v \times np}$ for $v \in V$. Each block then defined as $H_v = (H_{v1}, H_{v2}, \ldots, H_{vn})$ with, for $i = 1, \ldots, n$, the matrix $H_{vi} = A_v$ if node $i$ is included on the path going from node $v$ to the root node 1 i.e. $i \in \pi(v)$, and 0 otherwise.

The signal associated to the reformulated problem (11) remains sparse and is supported on a set $S$ with a particular structure due to encoding the sparsity of the differences $\{\Delta_e^\star\}_{e \in E}$. Specifically, the set $S$ contains the entries from $\{1, \ldots, p\}$ aligned with $S_1$ and, labeling the edges $e \in E$ with the integers $i = 1, \ldots, |E|$, the elements from $\{1, \ldots, p\}$ associated to $S_e$ offset by $i \times p$. Now that (1) is in terms of a Basis Pursuit problem, its success relies on the matrix $A$ satisfying the Restricted Null Space Property (10) with respect to the sparsity set $S$. This can be rewritten in terms of $x_1$ and $\{\Delta_e\}_{e \in E}$ as follows

$$\|(x_1)_{S_1}\|_1 + \sum_{e \in E} \|(\Delta_e)_{S_e}\|_1 \leq \frac{1}{2}\Big(\|x_1\|_1 + \sum_{e \in E} \|\Delta_e\|_1\Big) \tag{12}$$

$$\text{for } x_1 + \sum_{e \in \pi(v)} \Delta_e \in \mathrm{Ker}(A_v)\backslash\{0\} \text{ for } v \in V.$$

From now on we will let $S_1$ denote the largest $s$ entries of $x_1$, and for $e \in E$ the set $S_e$ as the largest $s'$ entries of $\Delta_e$. We that we begin with the proof of Theorem 4 in Section D.4, as the analysis of matching non-root matrices $A_v = A_w$ for $v, w \neq 1$ is simpler. This will then be followed by the proof of Theorem 3 in Section D.5.

## D.4 Proof of Theorem 4

We now provide the proof of Theorem 4. We begin with the following lemma which follows a standard shelling technique, see for instance [17].

**Lemma 2.** *Suppose the matrix $B \in \mathbb{R}^{N \times d}$ satisfies Restricted Isometry Property up-to sparsity level $d \geq k > 0$ with constant $\delta_k \in [0, 1)$. If $x \in \mathrm{Ker}(B)\backslash\{0\}$ then for any $U \subseteq \{1, \ldots, d\}$ such that $|U| = k$ we have*

$$\|(x)_U\|_2 \leq \frac{\delta_{2k}}{1 - \delta_k} \frac{1}{\sqrt{k}} \|x\|_1$$

*Proof of Lemma 2.* Noting that $Bx_U = -Bx_{U^c}$ and using the restricted isometry property of $B$, we can bound

$$(1 - \delta_k)\|(x)_U\|_2^2 \leq \|B(x)_U\|_2^2 = -\langle B(x)_U, B(x)_{U^c}\rangle.$$

Now decompose $U^c$ into disjoint sets $\{U_j\}_{j=1,2,\ldots}$ of size $k$ so that $U^C = U_1 \cup U_2 \cup \ldots$ . Structure the sets so that $U_1$ is the largest $k$ entries of $(x)_{U^c}$, $U_2$ is the largest $k$ entries of $(x)_{(U \cup U_1)^c}$ and so on. Note that for $j = 2, \ldots$, that $\|(x)_{B_j}\|_2 \leq \sqrt{k}\|(x)_{B_j}\|_\infty \leq \frac{1}{\sqrt{k}}\|(x)_{B_{j-1}}\|_1$. While $\|(x)_{B_1}\|_2 \leq \frac{1}{\sqrt{k}}\|x_U\|_1$. Returning to the equation above this allows us to bound with Proposition 1,

$$(1 - \delta_k)\|(x)_U\|_2^2 \leq \delta_{2k}\|(x)_U\|_2 \sum_{j \geq 1} \|(x)_{B_j}\|_2$$

$$\leq \frac{\delta_{2k}}{\sqrt{k}}\|(x)_U\|_2 \big(\|(x)_U\|_1 + \sum_{j \geq 1} \|(x)_{B_{j-1}}\|_1\big)$$

$$= \frac{\delta_{2k}}{\sqrt{k}}\|(x)_U\|_2 \|x\|_1$$

Dividing both sides by $(1 - \delta_k)\|(x)_U\|_2$ yields the result. $\qquad\square$

We now proceed to the proof of Theorem 4.

*Proof of Theorem 4.* Recall that it is sufficient to demonstrate that the restricted null space property for the reformulated problem (12) is satisfied with high probability. In this case we then have $A_v = \frac{1}{\sqrt{N_{\text{Non-root}}}} A_{\text{Non-Root}}$ for $v \in V \backslash \{0\}$, and $A_1 = \frac{1}{\sqrt{N_{\text{Root}}}} A_{\text{Root}}$. We then begin by assuming that $\frac{1}{\sqrt{N_{\text{Non-root}}}} A_{\text{Non-Root}}$ satisfies Restricted Isometry Property up-to sparsity level $k'$ with constant $\delta_{k'}^{\text{Non-root}} \in [0,1)$ and $\frac{1}{\sqrt{N_{\text{Root}}}} A_{\text{Root}}$ satisfies Restricted Isometry Property up-to sparsity level $k$ with constant $\delta_k^{\text{Root}} \in [0,1)$. Let us also suppose that $k \geq k'$. We will then return to satisfying this condition with high-probability at the end of the proof.

The proof proceeds by bounding $\|(x_1)_{S_1}\|_1, \|(\Delta)_{S_e}\|_1$ by using that $x = (x_1, \Delta_1, \ldots, \Delta_{|E|}) \in \text{Ker}(A) \backslash \{0\}$. We split into three cases: the root note $\|(x_1)_{S_1}\|_1$; the term $\|(\Delta)_{S_e}\|_1$ for edges $e = (v, w)$ not directly connected to the root $v, w \neq 1$; and the term $\|(\Delta)_{S_e}\|_1$ for edges $e = (v, w)$ joined to the root $v = 1$ or $w = 1$. Each is now considered in its own paragraph, with the combination in a fourth paragraph.

**Root Node** Note that $x_1 \in \text{Ker}(A_{\text{Root}}) \backslash \{0\}$ therefore from Lemma 2 and the inequality $\|(x_1)_{S_1}\|_1 \leq \sqrt{s}\|(x_1)_{S_1}\|_2$ we get the upper bound

$$\|(x_1)_{S_1}\|_1 \leq \frac{\delta_{2s}^{\text{Root}}}{1 - \delta_s^{\text{Root}}}\|x_1\|_1,$$

as required.

**Edges not connected to the root** For any edge $\widetilde{e} = (v, w) \in E$ not connected to the root node so $v, w \neq 1$, note we have $\Delta_{\widetilde{e}} \in \text{Ker}(A_{\text{Non-root}}) \backslash \{0\}$. To see this, each vector is in the same null-space $x_1 + \sum_{e \in \pi(v)} \Delta_e, x_1 + \sum_{e \in \pi(w)} \Delta_e \in \text{Ker}(A_{\text{Non-root}}) \backslash \{0\}$ so their difference is also. That is if $w$ is the furthest from the root node we can write $x_1 + \sum_{e \in \pi(w)} \Delta_e = x_1 + \sum_{e \in \pi(v)} \Delta_e + \Delta_{\widetilde{e}}$ and therefore

$$\Delta_{\widetilde{e}} = \left(x_1 + \sum_{e \in \pi(w)} \Delta_e\right) - \left(x_1 + \sum_{e \in \pi(v)} \Delta_e\right) \in \text{Ker}(A_{\text{Non-root}}).$$

An identical calculation can be done for the case of when $v$ is furthest from the root node. Therefore Lemma 2 yields the upper bound

$$\|(\Delta_{\widetilde{e}})_{S_{\widetilde{e}}}\|_1 \leq \frac{\delta_{2s'}^{\text{Non-root}}}{1 - \delta_{s'}^{\text{Non-root}}}\|\Delta_{\widetilde{e}}\|_1,$$

as required.

**Edges connected to the root** For edges connecting the root node so $\widetilde{e} = (v, w) \in E$ such that $v = 1$ or $w = 1$, begin by adding and subtracting $(x_1)_{S_{\widetilde{e}}}$ to decompose

$$\|(\Delta_{\widetilde{e}})_{S_{\widetilde{e}}}\|_1 \leq \|(x_1)_{S_{\widetilde{e}}}\|_1 + \|(x_1 + \Delta_{\widetilde{e}})_{S_{\widetilde{e}}}\|_1.$$

Now if $|S_{\widetilde{e}}| \leq k'$ we have from Lemma 2 the inequality $\|(x_1)_{S_{\widetilde{e}}}\|_1 \leq \sqrt{k'}\|x_{S_{\widetilde{e}}}\|_2 \leq \frac{\delta_{2k'}^{\text{Root}}}{1 - \delta_{k'}^{\text{Root}}}\|x_1\|_1$ since $x_1 \in \text{Ker}(A_{\text{Root}}) \backslash \{0\}$. As well as from the fact that $x_1 + \Delta_{\widetilde{e}} \in \text{Ker}(A_{\text{Non-root}}) \backslash \{0\}$ the upper bound $\|(x_1 + \Delta_{\widetilde{e}})_{S_{\widetilde{e}}}\|_1 \leq \frac{\delta_{2k'}^{\text{Non-root}}}{1 - \delta_{k'}^{\text{Non-root}}}\|x_1 + \Delta_{\widetilde{e}}\|_1$. Combining these bounds we get

$$\|(\Delta_{\widetilde{e}})_{S_{\widetilde{e}}}\|_1 \leq \frac{\delta_{2k'}^{\text{Root}}}{1 - \delta_{k'}^{\text{Root}}}\|x_1\|_1 + \frac{\delta_{2k'}^{\text{Non-root}}}{1 - \delta_{k'}^{\text{Non-root}}}\|x_1 + \Delta_{\widetilde{e}}\|_1$$

$$\leq \left(\frac{\delta_{2k'}^{\text{Root}}}{1 - \delta_{k'}^{Root}} + \frac{\delta_{2k'}^{\text{Non-root}}}{1 - \delta_{k'}^{\text{Non-root}}}\right)\|x_1\|_1 + \frac{\delta_{2k'}^{\text{Non-root}}}{1 - \delta_{k'}^{\text{Non-root}}}\|\Delta_{\widetilde{e}}\|_1$$

**Combining the upper bounds** Let us now combine the upper bounds for $\|(x_1)_{S_1}\|_1$ and $\|(\Delta_{\widetilde{e}})_{S_e}\|_1$ with $e \in E$. Summing them up and noting that there are $\text{Deg}(1)$ edges connecting the root yields

$$\|(x_1)_{S_1}\|_1 + \sum_{e \in E} \|(\Delta_e)_{S_e}\|_1 \leq \underbrace{\left(\frac{\delta_{2k}^{\text{Root}}}{1 - \delta_k^{Root}} + \text{Deg}(1)\frac{\delta_{2k'}^{\text{Root}}}{1 - \delta_{k'}^{Root}} + \text{Deg}(1)\frac{\delta_{2k'}^{\text{Non-root}}}{1 - \delta_{k'}^{\text{Non-root}}}\right)}_{\textbf{Multiplicative Term}}$$

$$\times \left(\|x_1\|_1 + \sum_{e \in E} \|\Delta_e\|_1\right)$$

We then require **Multiplicative Term** $\leq 1/2$ for the restricted Null space condition to be satisfied. Now, since $\delta_{2k}^{\text{Root}} \geq \delta_k^{\text{Root}}$ and $\delta_{2k'}^{\text{Non-root}} \geq \delta_{k'}^{\text{Non-root}}$, it is then sufficient for the Restricted Isometry Constants to satisfy the upper bounds

$$\delta_{2s}^{\text{Root}} \leq \frac{1}{3}$$

$$\delta_{2s'}^{\text{Root}} \leq \frac{1}{2\text{Deg}(1)}$$

$$\delta_{2s'}^{\text{Non-root}} \leq \frac{1}{2\text{Deg}(1)}$$

Leveraging Theorem 5 and taking a union bound, this is then satisfied with probability greater than $1 - \epsilon$ when

$$N_{\text{Root}} \geq 18C\big(\max\{s, \text{Deg}(1)^2 s'\}\log(ed) + \text{Deg}(1)^2 \log(1/\epsilon)\big),$$

$$N_{\text{Non-root}} \geq 18C\text{Deg}(1)^2\big(s'\log(ed) + \log(1/\epsilon)\big).$$

This concludes the proof. $\qquad\square$

## D.5 Proof of Theorem 3

We now present the proof of Theorem 3.

*Proof of Theorem 3.* Once again, recall it is sufficient to demonstrate the Restricted Null Space Property (11) is satisfied in this case. Following the proof of theorem 4, let the restricted isometry constant of $A_1$ up-to sparsity level $k$ be denoted $\delta_k^{\text{Root}} \in [0, 1)$. Meanwhile, let $\delta_{k'}^{\text{Non-root}}$ now denote the *maximum* restricted isometry constant of up-to sparsity level $k'$ of the matrices associated to non-root agents i.e. $\{A_v\}_{v \in \backslash\{1\}}$. Furthermore, let $\widetilde{A}_{\text{Combined}} \in \mathbb{R}^{(n-1)N_{\text{Non-root}} \times d}$ be constructed from the row-wise concatenation of the non-root agent matrices $\{A_v\}_{v \in V \backslash\{1\}}$. Similarly, let $\delta_{\widetilde{k}}^{\text{Combined}} \in [0, 1)$ denote the restricted isometry constant of $A_{\text{Combined}} := \widetilde{A}_{\text{Combined}}/\sqrt{n-1}$ up to sparsity level $\widetilde{k}$.

Following the proof of Theorem 4 we leverage that $x = (x_1, \Delta_1, \dots, \Delta_{|E|}) \in \text{Ker}(A)\backslash\{0\}$ to upper bound $\|(x_1)_{S_1}\|_1, \|(\Delta_e)\|_1$ for $e \in E$. In particular, we consider have three paragraphs: one associated to bounding $\|(x_1)_{S_1}\|_1$; one for bounding $\|(\Delta_e)\|_1$ for edges $e = (v, w) \in E$ not connected to the root $v, w \neq 1$; and one for bounding $\|(\Delta_e)\|_1$ for edges $e = (v, w)$ joined to the root $v = 1$ or $w = 1$. The fourth paragraph will then combined these bounds.

**Root Node** Since $x_1 \in \text{Ker}(A_1)$ we immediately have from Lemma 2 the upper bound for any $U \subset \{1, \dots, d\}$ such that $|U| = k$

$$\|(x_1)_U\|_2 \leq \frac{\delta_{2k}^{\text{Root}}}{1 - \delta_k^{\text{Root}}} \frac{1}{\sqrt{k}}\|x_1\|_1. \tag{13}$$

Setting $U = S_1$ and recalling and following the proof of Theorem 4, this immediately bounds $\|(x)_{S_1}\|_1 \leq \frac{\delta_{2s}^{\text{Root}}}{1 - \delta_s^{\text{Root}}}\|x_1\|_1$.

**Edges not connect to the root** Consider any edge $\widetilde{e} = (v, w) \in E$ not directly connected to the root i.e. $v, w \neq 1$. Without loss in generality suppose $w$ is the furthest from the root. This allows us to rewrite in terms of the difference

$$\Delta_{\widetilde{e}} = \Big(\sum_{e \in \pi(w)} \Delta_e\Big) - \Big(\sum_{e \in \pi(v)} \Delta_e\Big).$$

Each of the vectors are in potentially different null-spaces, and therefore, we bound each separately. Applying triangle inequality we then get

$$\|(\Delta_{\widetilde{e}})_{S_{\widetilde{e}}}\|_1 \leq \Big\|\Big(\sum_{e \in \pi(w)} \Delta_e\Big)_{S_{\widetilde{e}}}\Big\|_1 + \Big\|\Big(\sum_{e \in \pi(v)} \Delta_e\Big)_{S_{\widetilde{e}}}\Big\|_1$$

$$\leq \Big\|\Big(\sum_{e \in \pi(w)} \Delta_e\Big)_{U_1}\Big\|_1 + \Big\|\Big(\sum_{e \in \pi(v)} \Delta_e\Big)_{U_2}\Big\|_1$$

where $U_1, U_2$ are the largest $s'$ entries of $\sum_{e\in\pi(w)}\Delta_e$ and $\sum_{e\in\pi(v)}\Delta_e$ respectively. For the first term we have $x_1 + \sum_{e\in\pi(w)}\Delta_e \in \mathrm{Ker}(A_w)$ and therefore

$$(1-\delta_{k'}^{\text{Non-root}})\Big\|\Big(\sum_{e\in\pi(w)}\Delta_e\Big)_{U_1}\Big\|_2^2 \le \Big\|A_w\Big(\sum_{e\in\pi(w)}\Delta_e\Big)_{U_1}\Big\|_2^2$$

$$= -\Big\langle A_w\Big(\sum_{e\in\pi(w)}\Delta_e\Big)_{U_1}, A_w\Big(\sum_{e\in\pi(w)}\Delta_e\Big)_{U_1^c} + A_w x_1\Big\rangle$$

where we note that $A_w(x_1 + \sum_{e\in\pi(w)}\Delta_e) = 0$ and therefore $A_w\Big(\sum_{e\in\pi(w)}\Delta_e\Big)_{U_1} = -A_w\Big(\sum_{e\in\pi(w)}\Delta_e\Big)_{U_1^c} - A_w x_1$. Following the shelling argument in the proof of Lemma 2 we can upper bound

$$\Big|\Big\langle A_w\Big(\sum_{e\in\pi(w)}\Delta_e\Big)_{U_1}, A_w\Big(\sum_{e\in\pi(w)}\Delta_e\Big)_{U_1^c}\Big\rangle\Big| \le \frac{\delta_{2s'}^{\text{Non-root}}}{\sqrt{s'}}\Big\|\Big(\sum_{e\in\pi(w)}\Delta_e\Big)_{U_1}\Big\|_2\Big\|\sum_{e\in\pi(w)}\Delta_e\Big\|_1$$

Upper bound the other inner product as

$$\Big|\Big\langle A_w\Big(\sum_{e\in\pi(w)}\Delta_e\Big)_{U_1}, A_w x_1\Big\rangle\Big| \le \Big\|A_w\Big(\sum_{e\in\pi(w)}\Delta_e\Big)_{U_1}\Big\|_2\|A_w x_1\|_2$$

$$\le \sqrt{1+\delta_{s'}^{\text{Non-root}}}\Big\|\Big(\sum_{e\in\pi(w)}\Delta_e\Big)_{U_1}\Big\|_2\|A_w x_1\|_2$$

and dividing both sides by $(1-\delta_{s'}^{\text{Non-root}})\Big\|\Big(\sum_{e\in\pi(w)}\Delta_e\Big)_{U_1}\Big\|_2$ then yields

$$\Big\|\Big(\sum_{e\in\pi(w)}\Delta_e\Big)_{U_1}\Big\|_2 \le \frac{\delta_{2s'}^{\text{Non-root}}}{\sqrt{s'}}\Big\|\sum_{e\in\pi(w)}\Delta_e\Big\|_1 + \frac{\sqrt{1+\delta_{s'}^{\text{Non-root}}}}{1-\delta_{s'}^{\text{Non-root}}}\|A_w x_1\|_2$$

Repeating the steps above for the other node $v$ and going to $\ell_1$ norm from $\ell_2$ norm and bringing together the two bounds yields

$$\|(\Delta_{\widetilde{e}})_{S_{\widetilde{e}}}\|_1 \le \frac{\delta_{2s'}^{\text{Non-root}}}{1-\delta_{s'}^{\text{Non-root}}}\Big(\Big\|\sum_{e\in\pi(w)}\Delta_e\Big\|_1 + \Big\|\sum_{e\in\pi(v)}\Delta_e\Big\|_1\Big) \tag{14}$$

$$+ \frac{\sqrt{s'}\sqrt{1+\delta_{s'}^{\text{Non-root}}}}{1-\delta_{s'}^{\text{Non-root}}}\big(\|A_v x_1\|_2 + \|A_w x_1\|_2\big)$$

**Edges connecting to the root node** Consider an edge connected to the root node, that is, $\widetilde{e} = (v,w) \in E$ such that $v = 1$ or $w = 1$. Without loss in generality, let us suppose that $w = 1$. We can then bound using Restricted Isometry Property

$$(1-\delta_{s'})^{\text{Non-root}}\|(\Delta_{\widetilde{e}})_{S_{\widetilde{e}}}\|_2^2 \le \|A_v(\Delta_{\widetilde{e}})_{S_{\widetilde{e}}}\|_2^2$$

$$= -\langle A_v(\Delta_{\widetilde{e}})_{S_{\widetilde{e}}}, A_v(\Delta_{\widetilde{e}})_{S_{\widetilde{e}}^c} + A_v x_1\rangle$$

$$\le \frac{\delta_{2s'}^{\text{Non-root}}}{\sqrt{s'}}\|(\Delta_{\widetilde{e}})_{S_{\widetilde{e}}}\|_2\|\Delta_{\widetilde{e}}\|_1 + \|A_v(\Delta_{\widetilde{e}})_{S_{\widetilde{e}}}\|_2\|A_v x_1\|_2$$

$$\le \frac{\delta_{2s'}^{\text{Non-root}}}{\sqrt{s'}}\|(\Delta_{\widetilde{e}})_{S_{\widetilde{e}}}\|_2\|\Delta_{\widetilde{e}}\|_1 + \sqrt{1+\delta_{s'}^{\text{Non-root}}}\|(\Delta_{\widetilde{e}})_{S_{\widetilde{e}}}\|_2\|A_v x_1\|_2$$

where for the equality we note that $x_1 + \Delta_{\widetilde{e}} \in \mathrm{Ker}(A_v)\backslash\{0\}$ and therefore $A_v(\Delta_{\widetilde{e}})_{S_{\widetilde{e}}} = -A_v(\Delta_{\widetilde{e}})_{S_{\widetilde{e}}^c} - A_v x_1$. Meanwhile for the second inequality used a similar argument to previously. Dividing both sides by $(1-\delta_{s'}^{\text{Non-root}})\|(\Delta_{\widetilde{e}})_{S_{\widetilde{e}}}\|_2$ and going to $\ell_1$ norm we then get

$$\|(\Delta_{\widetilde{e}})_{S_{\widetilde{e}}}\|_1 \le \frac{\delta_{2s'}^{\text{Non-root}}}{1-\delta_{s'}^{\text{Non-root}}}\|\Delta_{\widetilde{e}}\|_1 + \frac{\sqrt{s'}\sqrt{1+\delta_{s'}^{\text{Non-root}}}}{1-\delta_{s'}^{\text{Non-root}}}\|A_v x_1\|_2 \tag{15}$$

**Combining upper bounds** Let us now combine the bounds on $\|(x)_{S_1}\|_1$ from (13), as well as the bounds (14) and (15) for the edges $e \in E$. This yields

$$\|(x_1)_{S_1}\|_1 + \sum_{e \in E} \|(\Delta)_{S_e}\|_1$$

$$= \|(x_1)_{S_1}\|_1 + \sum_{e=(v,w)\in E: v,w\neq 1} \|(\Delta_e)_{S_e}\|_1 + \sum_{e=(v,w)\in E: v=1 \text{ or } w=1} \|(\Delta_e)_{S_e}\|_1$$

$$\leq \frac{\delta_{2s}^{\text{Root}}}{1 - \delta_s^{\text{Root}}} \|x_1\|_1$$

$$+ \frac{\delta_{2s'}^{\text{Non-root}}}{1 - \delta_{s'}^{\text{Non-root}}} \underbrace{\sum_{e=(v,w)\in E: v,w\neq 1} \left( \Big\| \sum_{e\in\pi(w)} \Delta_e \Big\|_1 + \Big\| \sum_{e\in\pi(v)} \Delta_e \Big\|_1 \right)}_{\textbf{Term 1}}$$

$$\frac{\delta_{2s'}^{\text{Non-root}}}{1 - \delta_{s'}^{\text{Non-root}}} \sum_{e\in E: v=1 \text{ or } w=1} \|\Delta_e\|_1 + \underbrace{\frac{2\sqrt{s'}\sqrt{1+\delta_{s'}^{\text{Non-root}}}}{1 - \delta_{s'}^{\text{Non-root}}} \sum_{v\in V\setminus\{1\}} \|A_v x_1\|_2}_{\textbf{Term 2}}.$$

Where we must now bound **Term 1** and **Term 2**. To bound **Term 1** we simply apply triangle inequality to get

$$\textbf{Term 1} \leq \sum_{e=(v,w)\in E: v,w\neq 1} 2 \sum_{\widetilde{e}\in\pi(v)\cup\pi(w)} \|\Delta_{\widetilde{e}}\|_1 \leq 2\text{Deg}(V\setminus\{1\})\text{Diam}(G) \sum_{e\in E} \|\Delta_e\|_1$$

where $\pi(v) \cup \pi(w)$ denotes the union without duplicates. We then note that the sum $\sum_{e=(v,w)\in E: v,w\neq 1} 2\sum_{\widetilde{e}\in\pi(v)\cup\pi(w)} \cdots$ can be seen as counting the number of times an edge is used on a path from any non-root node to the root. The edges which appear on most paths to the root are those directly connected to the root. The number of edges feeding into the edge directly connected to the root is then upper bounded by the max degree of non-root nodes times the graph diameter $\text{Deg}(V\setminus\{1\})\text{Diam}(G)$. To bound **Term 2** we use Cauchy-Schwartz and recall the definition of $A_{\text{Combined}}$ to get

$$\sum_{v\in V\setminus\{v\}} \|A_v x_1\|_2 \leq \sqrt{n-1} \sqrt{\sum_{v\in V\setminus\{v\}} \|A_v x_1\|_2^2} = (n-1)\|A_{\text{Combined}} x_1\|_2$$

Using the fact that $x_1 \in \text{Ker}(A_1)\setminus\{0\}$ as well as the Restricted Isometry Property of $A_{\text{Combined}}$, we have show the following upper bound for $\ell \geq 1$

$$\textbf{Term 2} \leq \frac{6\sqrt{(1+\delta_{s'}^{\text{Non-root}})(1+\delta_{\ell}^{\text{Combined}})}}{(1-\delta_{s'}^{\text{Non-root}})(1-\delta_{2\ell}^{\text{Root}})} \frac{(n-1)\sqrt{s'}}{\sqrt{\ell}}) \|x_1\|_1 \tag{16}$$

The proof of (16) is then provided at the end. Bringing everything together and collecting constants we get

$$\|(x_1)_{S_1}\|_1 + \sum_{e\in E} \|(\Delta)_{S_e}\|_1$$

$$\leq 3\max \underbrace{\left\{ \frac{\delta_{2s}^{\text{Root}}}{1-\delta_s^{\text{Root}}}, \frac{2\text{Deg}(V\setminus\{1\})\text{Diam}(G)\delta_{2s'}^{\text{Non-root}}}{1-\delta_{s'}^{\text{Non-root}}}, \frac{6\sqrt{(1+\delta_{s'}^{\text{Non-root}})(1+\delta_{\ell}^{\text{Combined}})}}{(1-\delta_{s'}^{\text{Non-root}})(1-\delta_{2\ell}^{\text{Root}})} \frac{(n-1)\sqrt{s'}}{\sqrt{\ell}}) \right\}}_{\text{Multiplicative Term}}$$

$$\times \left( \|x_1\|_1 + \sum_{e\in E} \|\Delta\|_1 \right)$$

For the restricted nullspace property to be satisfied we must then ensure that **Multiplicative Term** $\leq 1/2$. This can then be ensured when setting $\ell = 156^2(n-1)^2 s'$ when the Restricted Isometry

constants satisfy

$$\delta_{2s}^{\text{Root}} \leq \frac{1}{4}$$

$$\delta_{2s'}^{\text{Non-root}} \leq \frac{1}{1 + 12D}$$

$$\delta_{2\ell}^{\text{Root}} \leq 1/2$$

$$\delta_{\ell}^{\text{Combined}} \leq 1$$

Using theorem 5, the conditions on $\delta_{2s}^{\text{Root}}, \delta_{2\ell}^{\text{Root}}$ and $\delta_{2s'}^{\text{Non-root}}$ are ensured with probability greater than $1 - \epsilon$ when

$$N_{\text{Root}} \geq C \times 32 \times 156^2 \max\{n^2 s', s\} \big( \log(ed) + \log(1/\epsilon) \big)$$

$$N_{\text{Non-root}} \geq C \times 13^2 \times \text{Deg}(V \backslash \{1\})^2 \text{Diam}(G)^2 s' \big( \log(ed) + \log(n/\epsilon) \big)$$

where $\delta_{2s'}^{\text{Non-root}}$ is the maximum restricted Isometry constant across the matrices $\{A_v\}_{v \in V \backslash \{1\}}$ and therefore, a union bound was taken. Meanwhile, for $\delta_{\ell}^{\text{Combined}}$, recall that the entries of $\sqrt{N_{\text{Non-root}}(n-1)} \times A_{\text{Combined}}$ are independent and sub-Gaussian i.e. $\widetilde{A}_{\text{Combined}}$ is the row-wise concatenation of $A_v = \widetilde{A}_v / \sqrt{N_{\text{Non-root}}}$ for $v \in V \backslash \{1\}$ where $\widetilde{A}_v$ has independent and identical sub-Gaussian entries. Therefore, following Theorem 5 the condition on $\delta_{\ell}^{\text{Combined}}$ is then satisfied when

$$N_{\text{Non-root}}(n-1) \geq C \times 156^2 (n-1)^2 s' \big( \log(ed) + \log(1/\epsilon) \big)$$

Dividing both sides by $n - 1$ and combining the conditions on $N_{\text{Non-root}}$ yields the result.

Let us now prove (16). Using (13) with $U$ being the largest $\ell$ entries of $x_1$ we have (since $x_1 \in \text{Ker}(A_1) \backslash \{0\}$)

$$\|x_1\|_2 \leq \|(x_1)_{U^c}\|_2 + \|(x_1)_U\|_2 \leq \frac{1}{\sqrt{\ell}} \|x_1\|_1 + \frac{\delta_{2\ell}^{\text{Root}}}{1 - \delta_{\ell}^{\text{Root}}} \frac{1}{\sqrt{\ell}} \|x_1\|_1 = \Big(1 + \frac{\delta_{2\ell}^{\text{Root}}}{1 - \delta_{\ell}^{\text{Root}}}\Big) \frac{1}{\sqrt{\ell}} \|x_1\|_1$$

where have bounded using the shelling argument $U^c = B_1 \cup B_2 \cup \ldots$ as $\|(x)_{U^c}\|_2 \leq \sum_{j \geq 1} \|(x)_{B_j}\|_2 \leq \frac{1}{\sqrt{\ell}} \|x\|_1$. That is $B_1$ is the largest $\ell$ entries of $x_1$ in $U^c$, $B_2$ is the largest $\ell$ entries in $(U \cup B_1)^c$ and so on. We then have $\|(x)_{B_{j-1}}\|_2 \leq \sqrt{\ell} \|(x)_{B_{j-1}}\|_\infty \leq \frac{1}{\sqrt{\ell}} \|(x)_{B_j}\|_1$. Therefore we can bound with $c = \Big(1 + \frac{\delta_{2\ell}^{\text{Root}}}{1 - \delta_{\ell}^{\text{Root}}}\Big)$

$$\|A_{\text{Combined}} x_1\|_2 = \|A_{\text{Combined}} \frac{x_1}{\|x_1\|_1}\|_2 \|x_1\|_1 \leq \Big( \max_{x : \|x\|_2 \leq \frac{c}{\sqrt{\ell}}, \|x\|_1 \leq 1} \|A_{\text{Combined}} x\|_2 \Big) \|x_1\|_1$$

where if we fix $x = \frac{x_1}{\|x_1\|_1}$ then it is clear $\|x\|_1 = 1$ and $\|x\|_2 = \frac{\|x_1\|_2}{\|x_1\|_1} \leq \frac{c}{\sqrt{\ell}}$. We now study the maximum above. In particular, it can be rewritten since $c \geq 1$

$$\max_{x : \|x\|_2 \leq \frac{c}{\sqrt{\ell}}, \|x\|_1 \leq 1} \|A_{\text{Combined}} x\|_2 = \frac{c}{\sqrt{\ell}} \max_{x : \|x\|_2 \leq 1, \|x\|_1 \leq \frac{\sqrt{\ell}}{c}} \|A_{\text{Combined}} x\|_2$$

$$\leq \frac{c}{\sqrt{\ell}} \max_{x : \|x\|_2 \leq 1, \|x\|_1 \leq \sqrt{\ell}} \|A_{\text{Combined}} x\|_2$$

$$= \frac{c}{\sqrt{\ell}} \max_{x \in \mathbb{B}_2(1) \cap \mathbb{B}_1(\sqrt{\ell})} \|A_{\text{Combined}} x\|_2$$

Using Lemma 1 we can then bound

$$\max_{x \in \mathbb{B}_2(1) \cap \mathbb{B}_1(\sqrt{\ell})} \|A_{\text{Combined}} x\|_2 \leq \max_{x \in \mathbb{B}_2(3) \cap \mathbb{B}_0(\ell)} \|A_{\text{Combined}} x\|_2$$

$$\leq 3 \sqrt{1 + \delta_{\ell}^{\text{Combined}}}$$

where at the end used the Restricted Isometry Property of $A_{\text{Combined}}$. Bringing everything together we get the bound for **Term 2** with $c = 1 + \frac{\delta_{2\ell}^{\text{Root}}}{1 - \delta_{\ell}^{\text{Root}}} \leq \frac{1}{1 - \delta_{2\ell}^{\text{Root}}}$

$$\textbf{Term 2} \leq \frac{6\sqrt{(1 + \delta_{s'}^{\text{Non-root}})(1 + \delta_{\ell}^{\text{Combined}})}}{(1 - \delta_{s'}^{\text{Non-root}})(1 - \delta_{2\ell}^{\text{Root}})} \frac{(n-1)\sqrt{s'}}{\sqrt{\ell}}) \|x_1\|_1$$

as required.

$\square$

# E  Proofs for Noisy Case

In this section we provide the proofs for the noisy setting. Section E.1 begin by introducing the problem of Basis Pursuit Denoising. Section E.2 presents the proof of Theorem 2. Section E.3 presents the proof for an intermediate lemma.

## E.1  Basis Pursuit Denoising

Let us begin by introducing Basis Pursuit Denoising. That is suppose $y = Ax^\star + \epsilon$ for some noise $\epsilon \in \mathbb{R}^n$. The Basis Pursuit Denoising problem [9] then considers replacing the equality with a bound on the $\ell_2$. Namely for $\eta \geq 0$

$$\min_x \|x\|_1 \text{ subject to } \|Ax - y\|_2 \leq \eta. \tag{17}$$

Naturally, the equality constraint $Ax = y$ in the noiseless setting has been swapped for an upper bound on the discrepancy $\|Ax - y\|_2$. To investigate guarantees for the solution to (17), we consider the Robust Null Space Property, see for instance [17]. A matrix $A$ is said to satisfy the Robust Null Space Property for a set $S \subseteq \{1, \ldots, p\}$ and parameters $\rho, \tau \geq 0$ if

$$\|x_S\|_1 \leq \rho \|x_{S^c}\|_1 + \tau \|Ax\|_2 \text{ for all } x \in \mathbb{R}^N. \tag{18}$$

Given condition (18), bounds on the $\ell_1$ estimation error between a solution to the Denoising Basis Pursuit problem (17) and the true underlying signal $x^\star$ can be obtained. That is, for any solution to (17), $x \in \mathbb{R}^p$ with $y = Ax^\star + e$ where $\|e\|_2 \leq \eta$, we have (see [17] with $z = x^\star$)

$$\|x - x^\star\|_1 \leq \underbrace{\frac{2(1 + \rho)}{1 - \rho} \|(x^\star)_{S^c}\|_1}_{\text{Sparse Approximation}} + \underbrace{\frac{4\tau}{1 - \rho} \eta}_{\text{Noise}}.$$

The first term above encodes that $x^\star$ is not exactly $s$ sparse, while the second term represents error from the noise. We now discuss the values taken by $\eta$ and $\tau$ in the case that $A$ has i.i.d. sub-Gaussian entries. Recall Theorem 5 that the scaled matrix $A/\sqrt{N}$ in this case can satisfy a Restricted Isometry Property, and thus, it is natural to choose $\eta = \sqrt{N}\eta_{\text{Noise}}$ for $\eta_{\text{Noise}} \geq 0$ since the $\ell_2$ bound on the residuals in (17) becomes $\|Ax - y\|_2/\sqrt{N} \leq \eta_{\text{Noise}}$. We can then pick $\|e\|_2/\sqrt{N} \leq \eta_{\text{Noise}}$, which is an upper bound on the standard deviation of the noise. The Robust Null Space Property then holds in this case, see [17, Theorem 4.22], with $\tau \approx \sqrt{s}$, leading to a $\ell_1$ error bound of the order $\|x - x^\star\|_1 \lesssim \|(x^\star)_{S^c}\|_1 + \eta_{\text{Noise}}\sqrt{s}$ (see [49]).

## E.2  Proof of Theorem 2

We begin by recalling Section D.3 which reformulated the Total Variation Basis Pursuit problem (1) into a Basis Pursuit problem (9). In particular, we note that Total Variation Basis Pursuit Denoising (5) can be reformulated into Basis Pursuit Denoising (17) in a similar manner. Let us suppose the root node signal $x_1^\star$ and the $\{\Delta_e^\star\}_{e \in E}$ are approximately sparse and each agent $v \in V$ holds noisy samples $y_v \approx A_v x_v^\star$. Reformulating the Total Variation Basis Pursuit problem into a Basis Pursuit problem (11) and bounding the $\ell_2$ norm of the residuals, then yields the *Total Variation Basis Pursuit Denoising* problem

$$\min_{x_1, \Delta_e \in E} \|x_1\|_1 + \sum_{e \in E} \|\Delta_e\|_1 \text{ subject to} \tag{19}$$

$$\sum_{v \in V} \|A_v (x_1 + \sum_{e \in \pi(v)} \Delta_e) - y_v\|_2^2 \leq \eta^2.$$

Where $\eta^2$ now upper bounds the squared $\ell_2$ norm of the noise summed across all of the nodes i.e. $\sum_{v \in V} \|A_v x_v^\star - y_v\|_2^2$. This is now in the form of (17) with an augmented matrix $A$ as in the reformulated problem described in Section D.3.

Let us now recall that $\delta_k^{\text{Non-root}}$ denotes the largest Restricted Isometry Constant of the matrices associated to the non-root agents $\{A_v\}_{v \in V \setminus \{1\}}$. Similarly, $\delta_k^{\text{Root}}$ denotes the Restricted Isometry Constant associated to the root matrix $A_1$. The following theorem then gives, values for $\rho$ and $\tau$ for which the augmented matrix $A$ and sparsity set $S$ (described in Section D.3) satisfies the Robust Null Space Property (18).

**Lemma 3.** *Consider the $A$ matrix and sparsity set $S$ as constructed in Section D.3. Then $A$ satisfies the Robust Null Space Property with $\rho = \rho'/(1 - \rho')$ and $\tau = \tau'/(1 - \rho')$ where*

$$\rho' = 4\left(\frac{n\delta_{2s'}^{\text{Non-root}}}{1 - \delta_{s'}^{\text{Non-root}}} \vee \frac{\delta_{2s}^{\text{Root}}}{1 - \delta_s^{\text{Root}}}\right) \quad and$$

$$\tau' = \frac{\sqrt{1 + \delta_{s'}^{\text{Non-root}}}}{1 - \delta_{s'}^{\text{Non-root}}} \vee \frac{\sqrt{1 + \delta_s^{\text{Root}}}}{1 - \delta_s^{\text{Root}}}\left(\sqrt{s} + Deg(G)\sqrt{ns'}\right).$$

Note the parameter $\tau'$ scales (up to a network degree Deg(G) factor) with the sparsity of the differences. Naturally we require $\rho' < 1/2$, which can be ensured if each agent $\{A_v\}_{v \in V}$ have i.i.d. sub-Gaussian matrices. In particular, we require

$$\delta_{2s}^{\text{Root}} \leq \frac{1}{9} \text{ and } \delta_{2s'}^{\text{Non-root}} \leq \frac{1}{1 + 8n}$$

Following Theorem 5 this can be ensured with probability greater than $1 - \epsilon$ when

$$N_{\text{Root}} \geq 81C\big(2s\log(ed/s) + \log(n/\epsilon)\big)$$

$$N_{\text{Non-root}} \geq 81n^2C\big(2s'\log(ed/s') + \log(n/\epsilon)\big)$$

Choosing $\eta = \sqrt{\sum_{v \in V} N_v}\eta_{\text{Noise}}$ where $\eta_{\text{Noise}} > 0$ upper bounds the noise standard deviation across all of the agents, the $\ell_1$ estimation error of the solution to (5) is then of the order

$$\|x_1 - x_1^\star\|_1 + \sum_{e \in E} \|\Delta_e - \Delta_e^\star\|_1 \lesssim \underbrace{\|(x^\star)_{S^c}\|_1}_{\text{Approximate Sparsity}} + \underbrace{(\sqrt{s} + Deg(G)\sqrt{ns'})\eta_{\text{Noise}}}_{\text{Noise}}.$$

The error scales with the approximate sparsity of the true signal through $\|(x^\star)_{S^c}\|_1$ and now the noise term with the effective sparsity $\sqrt{s} + Deg(G)\sqrt{ns'}$. Picking $x^\star$ to be supported on $S$, the approximate sparsity term goes to zero, as required.

### E.3   Proof of Lemma 3

We now set to show that the Robust Null Space Property (18) holds for some $\rho, \tau$. We note it suffices to show the following which is equivalent to the Robust Null Space Property

$$\|(x)_S\|_1 \leq \rho'\|x\|_1 + \tau'\|Ax\|_2 \text{ for all } x \in \mathbb{R}^N.$$

In particular, by adding $\rho\|(x)_S\|_1$ to both sides of the inequality for the Robust Null Space Property (18) and dividing by $1 + \rho$, we see that if the above holds then the Robust Null Space Property holds with $\rho = \frac{\rho'}{1 - \rho'}$ and $\tau = \tau'/(1 - \rho')$.

The proof naturally follows the noiseless case (proof for Theorem 3) although with an additional error term owing the noise. To make the analysis clearer, the steps following the noiseless case from the proof of Theorem 3, are simplified.

We begin by controlling the $\ell_1$ norm of $x_1 + \sum_{e \in \pi(v)} \Delta_e = x_v$ for $v \in V$ restricted to subsets $U$. Considering the subset $U$ of size $|U| \leq s'$, and in particular, the set $U$ associated to the largest $s'$ entries of $x_v$. Following the shelling argument used within the proof of Lemma 2 decompose $U^c = B_1 \cup B_2 \cup \ldots$ where $B_1$ are the largest $s'$ entries of $x_v$ within $U^c$, $B_2$ are the $s'$ largest entries

of $x_v$ in $(U \cup B_1)^c$ and so on. We can then upper bound

$$(1 - \delta_{s'}^{\text{Non-Root}})\|(x_v)_U\|_2^2 \leq \|A_v(x_v)_U\|_2^2$$

$$= \Big\langle A_v(x_v)_U, A_v\Big(x_v - \sum_{j \geq 1}(x_v)_{B_j}\Big)\Big\rangle$$

$$= \langle A_v(x_v)_U, A_v x_v\rangle - \sum_{j \geq 1}\langle A_v(x_v)_U, A_v(x_v)_{B_j}\rangle$$

$$\leq \sqrt{1 + \delta_{s'}^{\text{Non-root}}}\|(x_v)_U\|_2\|A_v x_v\|_2 + \frac{\delta_{2s'}^{\text{Non-root}}}{\sqrt{s'}}\|(x_v)_U\|_2\|x_v\|_1$$

where we used the Restricted Isometry Property of $A_v$ to upper bound inner product $\langle A_v(x_v)_U, A_v x_v\rangle \leq \|A_v(x_v)_U\|_2\|A_v x_v\|_2 \leq \sqrt{1 + \delta_{s'}^{\text{Non-root}}}\|(x_v)_U\|_2\|A_v x_v\|_2$ and followed the steps in the proof of Lemma 2 to upper bound $\sum_{j \geq 1}\langle A_v(x_v)_U, A_v(x_v)_{B_j}\rangle \leq \delta_{2s'}^{\text{Non-root}}\|(x_v)_U\|_2\sum_{j \geq 1}\|(x_v)_{B_j}\|_1 \leq \frac{1}{\sqrt{s'}}\delta_{2s'}^{\text{Non-root}}\|(x_v)_U\|_2\|x_v\|_1$. Dividing both sides by $(1 - \delta_{s'}^{\text{Non-root}})\|(x_v)_U\|_2$ we then get

$$\|(x_v)_U\|_2 \leq \frac{\delta_{2s'}^{\text{Non-root}}}{1 - \delta_{s'}}\frac{1}{\sqrt{s'}}\|x_v\|_1 + \frac{\sqrt{1 + \delta_{s'}^{\text{Non-root}}}}{1 - \delta_{s'}^{\text{Non-root}}}\|A_v x_v\|_2$$

Using that $\|(x_v)_U\|_2 \geq \frac{1}{\sqrt{s'}}\|(x_v)_U\|_1$ as well as simply upper bounding $\|x_v\|_1 = \|x_1 + \sum_{e \in \pi(v)}\Delta_e\|_1 \leq \|x_1\|_1 + \sum_{e \in \pi(v)}\|\Delta_e\|_1 \leq \|x_1\|_1 + \sum_{e \in E}\|\Delta_e\|_1$ we have

$$\Big\|\Big(x_1 + \sum_{e \in \pi(v)}\Delta_e\Big)_U\Big\|_1 \leq \frac{\delta_{2s'}^{\text{Non-root}}}{1 - \delta_{s'}}\Big(\|x_1\|_1 + \sum_{e \in E}\|\Delta_e\|_1\Big) + \frac{\sqrt{1 + \delta_{s'}^{\text{Non-root}}}}{1 - \delta_{s'}^{\text{Non-root}}}\sqrt{s'}\|A_v x_v\|_2. \quad (20)$$

For $e = \{v, w\} \in E$ we now set to bound $\|(\Delta_e)_{S_e}\|_1$ where recall $S_e$ are the largest $s'$ elements of $\Delta_e$. Suppose $w$ is closest to the root node. If not, swap the $v, w$ in the following. By adding and subtracting $\big(x_1 + \sum_{\tilde{e} \in \pi(w)}\Delta_{\tilde{e}}\big)_{S_e}$ we then get

$$\|(\Delta_e)_{S_e}\|_1 \leq \Big\|\Big(x_1 + \sum_{\tilde{e} \in \pi(w)}\Delta_{\tilde{e}}\Big)_{S_e}\Big\|_1 + \Big\|\Big(x_1 + \sum_{\tilde{e} \in v}\Delta_{\tilde{e}}\Big)_{S_e}\Big\|_1$$

$$\leq \frac{2\delta_{2s'}^{\text{Non-root}}}{1 - \delta_{s'}}\Big(\|x_1\|_1 + \sum_{e \in E}\|\Delta_e\|_1\Big) + \frac{\sqrt{1 + \delta_{s'}^{\text{Non-root}}}}{1 - \delta_{s'}^{\text{Non-root}}}\sqrt{s'}\big(\|A_v x_v\|_2 + \|A_w x_w\|_2\big)$$

where on the second inequality we applied (20) twice. Summing the above over all edges $e \in E$, we note $\|A_v x_v\|_2$ for $v \in V$ appears at most the max degree of the graph, as such we get

$$\sum_{e \in E}\|(\Delta_e)_{S_e}\|_1 \leq \frac{2n\delta_{2s'}^{\text{Non-root}}}{1 - \delta_{s'}^{\text{Non-root}}}\Big(\|x_1\|_1 + \sum_{e \in E}\|\Delta_e\|_1\Big) + \frac{\sqrt{1 + \delta_{s'}^{\text{Non-root}}}}{1 - \delta_{s'}^{\text{Non-root}}}\text{Deg}(G)\sqrt{s'}\sum_{v \in V}\|A_v x_v\|_2$$

$$\leq \frac{2n\delta_{2s'}^{\text{Non-root}}}{1 - \delta_{s'}^{\text{Non-root}}}\Big(\|x_1\|_1 + \sum_{e \in E}\|\Delta_e\|_1\Big) + \frac{\sqrt{1 + \delta_{s'}^{\text{Non-root}}}}{1 - \delta_{s'}}\text{Deg}(G)\sqrt{ns'}\sqrt{\sum_{v \in V}\|A_v x_v\|_2^2}$$

where on the final inequality we upper bounded $\sum_{v \in V}\|A_v x_v\|_2 \leq \sqrt{n}\sqrt{\sum_{v \in V}\|A_v x_v\|_2^2}$.

We now consider the bound for $\|(x_1)_U\|_1$ but for subsets $U$ of size up to $s$. Following an identical set of steps as for (20), but with $s'$ swapped with $s$ and $\delta_{s'}$ swapped with $\delta_s^{(1)}$, we get the upper bound

$$\|(x_1)_U\|_1 \leq \frac{\delta_{2s}^{\text{Root}}}{1 - \delta_s^{\text{Root}}}\Big(\|x_1\|_1 + \sum_{e \in E}\|\Delta_e\|_1\Big) + \frac{\sqrt{1 + \delta_s^{\text{Root}}}}{1 - \delta_s^{\text{Root}}}\sqrt{s}\|A_1 x_1\|_2$$

$$\leq \frac{\delta_{2s}^{\text{Root}}}{1 - \delta_s^{\text{Root}}}\Big(\|x_1\|_1 + \sum_{e \in E}\|\Delta_e\|_1\Big) + \frac{\sqrt{1 + \delta_s^{\text{Root}}}}{1 - \delta_s^{\text{Root}}}\sqrt{s}\sum_{v \in V}\sqrt{\|A_v x_v\|_2^2}$$

where at the end we simply upper bounded $\|A_1 x_1\|_2 = \sqrt{\|A_1 x_1\|_2^2} \leq \sqrt{\sum_{v \in V}\|A_v x_v\|_2^2}$. Picking $U = S_1$, adding together the upper bound for $\sum_{e \in E}\|(\Delta_e)_{S_e}\|_1$ and $\|(x_1)_U\|_1$, and collecting terms then yields the result.