# OpenReview forum: "Distributed Machine Learning with Sparse Heterogeneous Data"
_NeurIPS.cc/2021/Conference — NeurIPS 2021 Poster_

### Official Review · Reviewer_MYEC · 2021-07-16

**Rating:** 6
**Confidence:** 4

**Summary:**

The authors introduce a method for learning signals on a graph, whereby each node signal x on the graph G to be learnt is detected by signals y via a linear transformation y = Ax. The method is known as total variation basis pursuit, and exploits the difference between signals connected by the edges of the graph to learn the signals.

**Limitations And Societal Impact:**

Yes

**Main Review:**

The contributions of the authors are twofold: One, they do not require the underlying graph to be known while achieving better sample complexity than the algorithms they compared to, which are independent basis pursuit (BP), stepwise BP and group lasso. Two, they do not require the incoherence conditions associated with support sets of the matrix. Instead, it is reformulated as a Restricted Null Space Property that holds when the sample size scales with the sparsity along the graph edges.

Generally the paper is written reasonably. However, I have some questions over the main result and theorem, which I think can help to make the paper clearer.

1) The optimization problem (equation 1) is formulated in terms of a tree graph. While the purpose of this algorithm is to tackle the case where the underlying graph is not known, why is there an assumption for the underlying graph to be a tree? Some discussion on this would be helpful. The idea of a root or tree is also discussed in the abstract in line 8 without prior definition or discussion as to why this case in considered in the first place, which may lead to confusion among readers.

2) I think more discussion on the limitations of the unknown graph G is important, since the optimization formulation does not take into account the sparsity of G. I fail to understand how different graph structures will affect the results of learning, and it seems like it should. If this is not important, I think the authors should discuss why it is not, especially since the learning is done on an underlying graph network.

3) Instead of the sketch of the proof, I think more discussion on the comparison between the TVBP and the other methods is more valuable. The authors only discuss the difference in sample complexity, whereas it may be more valuable to discuss the different formulations used in independent BP, stepwise BP and the group lasso/GSP model. Otherwise, readers new to the field are not able to distinguish what is novel in this formulation of the TVBP.

With these added to the paper, I think the paper can be accepted.

Some typos:
i) line 84 - v \in v should be v \in V.

**Time Spent Reviewing:**

10 hours

---

> ### Author Response · Authors · 2021-08-09
> **Comment to Reviewer Three**
>
> We provide answers to specific questions in the following bullet point format.
>
> * Purpose of considering tree graph in equation (1)
>     -  Algorithm (1) considers a tree topology as the proof both: maps (1) to a Basis Pursuit objective (see **Reformulating TVBP problem** in proof sketch), and uses the Restricted Nullspace Property.  Extending to a non-tree graph is challenging as it is not possible, to the best of our knowledge, to map (1) to an equivalent Basis Pursuit problem i.e. in terms of differences along edges. An alternative approach is to follow the robust null space analysis, with bounds then in terms of the norm penalty (which is stronger for a grid than a cycle, say, as there are more edges).
>
> * Affect of graph $G$ on learning.
>     - See **Adaptitvity to unkown graph** $G$ **and edge sparsity** $\mathbf{s}^{\prime}$ in Comment to all Reviewers.
>
> * Discussion on comparison between formulation of TVBP, Independent BP, Stepwise BP and group lasso
>     - Indeed, each of the methodologies listed start with different formulations to the joint recovery problem. To make this distinction clearer, section 2.1 (Setup) now includes the optimisation problems associated to each the methods alongside a discussion.
>
> * Some typos: i) line 84 - $v \in v$ should be $v \in V$
>     - Thank you noting this typo, this has now been updated.

---

> ### Author Response · Authors · 2021-09-06
> **Follow up to Reviewer Three**
>
> Dear Reviewer,
>
> Just a quick follow up regarding our response to your comments.
> Specifically, If you have any other outstanding concerns, do let us know as we can provide additional clarification.
>
> Otherwise, if you have no other concerns, we would kindly ask that you update your review in light of this.
>
> Best,
> Authors.

---

### Official Review · Reviewer_Uu6J · 2021-07-16

**Rating:** 6
**Confidence:** 4

**Summary:**

The paper develops and analyzes a total variation formulation to recover multiple signals associated with nodes in a graph under sparsity assumptions on the signals at each node and the difference signals along the edges. Sample complexity results are presented for the noiseless and noisy settings, and a distributed ADDM-based algorithm is developed.

**Main Review:**

Strength
* The method proposed could be better than Group Lasso when the overlap between the signals’ supports is small.
* The paper provides provable guarantees and is generally well-written.

Weakness
* The sample complexity seems to be generally worse than Group Lasso and the Dirty Model. One of the terms scales quadratically with the graph size n (vs linear in n for Group Lasso), which calls into question the scalability of the proposed method and its usefulness for large graphs. In turn, all examples in the numerical results are for fairly small graphs.
* The title of the paper is too broad and does not reflect the actual scope of this work. This is essentially an MMV recovery problem with sparsity constraints relative to a graph topology.
* The experimental results for hyperspectral imaging are not convincing and inconclusive.
* I am not able to draw conclusions about the actual gains of the proposed total variation method relative to existing methods. For example, it would have been useful to derive phase transitions type results to possibly highlight regimes where the proposed TVBP method outperforms Group Lasso as a function of the overlap between the signal supports.

Overall, I believe this is a good paper but is lacking in several aspects. Compared to other papers I have reviewed at NeurIPS, this paper is borderline.


**Time Spent Reviewing:**

3

---

> ### Author Response · Authors · 2021-08-09
> **Comment to Reviewer Two**
>
> We provide answers to specific questions in the following bullet point format.
>
> * Quadratic dependence on nodes $n$ and Multiple Measurement Vector Framework (MMV)
>     - The sample complexities provided in Table 1 are in a *more general* setting than the MMV framework, the latter of which assumes the sensing matrices at each node are *identical*. Table 2 (3rd row) provides the sample complexities for TVBP aligned with the MMV framework, which *does not* have the quadratic dependence on $n$. This arises from the null-spaces of the agents being the same, allowing the errors outside the support set to more easily controlled. This distinction is better clarified within the manuscript.
>
> * Experimental results for hyperspectral imaging
>      - Our work provides a *precise theoretical understanding* of the total variation penalty in sparse recovery, guiding the future development of hyperspectral (and other) methodologies. In this regard, there are many works dedicated to developing methodologies for the total variational penalty in hyperspectral imaging (see [23] and its citations), and thus, the experiments demonstrate that the  methodology we consider already yields qualitative improvements for a real world problem.

---

> ### Comment · Reviewer_Uu6J · 2021-08-31
> **After reading the authors' response:**
>
> My understanding is that the results for Group Lasso and the dirty model are also worst case and do not require the knowledge of the graph. They seem to yield significantly better sample complexities for simultaneous recovery of the sparse models without incorporating knowledge of the true graph. I am not able to see exactly what gains the proposed TVBP brings in the realistic and most interesting case where the true graph is unknown. On the other hand, it is not clear whether Group Lasso can also benefit from knowledge of the graph. Shouldn’t that be added to Table 2 for comparison? Also, there does not seem to be any savings in sample complexity for relatively dense graphs with small diameter. The work can greatly benefit from the inclusion of comparison results with phase transitions highlighting the gains, where experimental results are presented to verify the findings of the theoretical analysis.
>
> Other algorithms are also amenable to decentralized implementations, such as the decentralized Group Lasso, so this is by no means exclusive to the proposed approach. Also, the non-cited [Chen et al, JMLR 2010] considers scalable algorithms for different penalties for general structures, including the overlapping group lasso, fused lasso, and other graph-guided penalties, so I wonder how these relate and compare to the proposed approach?
>
> I strongly believe that the paper has merit but I continue to have the same questions and concerns about clarity of the presentation, clear discussion of the gains, and inconclusiveness of the experimental results.

---

> > ### Author Response · Authors · 2021-09-01
> > **Response to Uu6J Comment**
> >
> > We thank the reviewer for "strongly believe that the paper has merit" and giving the additional feedback. Following this feedback, we have provided some additional clarification below.
> >
> > + "*They [Group Lasso and the dirty model] seem to yield significantly better sample complexities for simultaneous recovery of the sparse models without incorporating knowledge of the true graph. I am not able to see exactly what gains the proposed TVBP brings in the realistic and most interesting case where the true graph is unknown.*"
> >     - To the best of knowledge, the Group Lasso and Dirty Model **do not** yield significantly better theoretical sample complexities in the most general case. As described in the Table 1 of the manuscript, it depends upon the structure of the signals, namely, their overlap along edges in the graph.
> >
> > + "*On the other hand, it is not clear whether Group Lasso can also benefit from knowledge of the graph.*"
> >     - As far as we are aware, guarantees for the group lasso rely on satisfying an incoherence condition (see references within manuscript), which requires the sample complexity at each node to scale with the support size. This would yield a total sample complexity of $O(n(s+s^{\prime}))$, where as the TVBP scales as $O(s + n^2 \text{Diam}(G) s^{\prime})$.
> >
> > + "*Other algorithms are also amenable to decentralized implementations, such as the decentralized Group Lasso...*"
> >     - Thank you for highlighting this avenue of research. The only paper we are aware of on the decentralized group lasso is ``Robust Group LASSO over Decentralized Networks`` Manxi Wang et. al. 2016, which albeit it describes a decentralized version of the group lasso, it does not establish any theoretical guarantees for it. In our work, we thought to mainly compare our methods to algorithms for which sample guarantees are known, c.f. Table 1. We will nevertheless include additional details on these within the related literature section.
> >
> > + "*Also, the non-cited [Chen et al, JMLR 2010] considers scalable algorithms for different penalties for general structures, including the overlapping group lasso, fused lasso, and other graph-guided penalties, so I wonder how these relate and compare to the proposed approach?*"
> >     - Could the reviewer please specify which manuscript [Chen et al., JMLR 2010] is referring to.
> >     - We note that our work focuses on providing precise theoretical guarantees for a joint recovery problem with a fundamentally different formulation to the group lasso and dirty model. As such, we feel it is more appropriate to compare the other primitives methods e.g. group lasso and dirty model, versus more tailored algorithms, such as those considered within the hyperspectral community.
> >
> > We would be happy to address any additional points that the reviewer might still find unclear. Thank you.

---

### Official Review · Reviewer_y67W · 2021-07-16

**Rating:** 6
**Confidence:** 3

**Summary:**

This paper studies distributed (linear model) learning on heterogeneous data, e.g. federated learning, with the assumption of sparsity. Given a siilarity graph, it minimizes the l1 norm of the sparse model at a root node as well as the total variation along the graph paths. This work provides statistical analyses on the sample complexity by reformulating the problem as a basis pursuit denoising problem, and show that sample complexity is improved compared to previous works. A distributed ADMM algorithm is derived, and synthetic data experiments and application to hyperspectral unmixing are demonstrated.

**Limitations And Societal Impact:**

"Specifically, the theorems can provide guidance on developing algorithms in the context of distributed machine learning and hyper-spectral imaging, which can have both negative and positive impacts." This is true, but it would be helpful to list a few of such potential impacts. For example, distributed machine learning has connections to privacy issues, and hyperspectral imaging to military operations.

**Main Review:**

This paper is clearly written and well presented, but it could benefit from connecting the theoretical results to the synthetic data experiment results more. For example, comparing the phase transition diagram in Figure 1 to the sample complexity derived in Theorem 1 would be a nice way to demonstrate the soundness of the paper's theoretical claims, and similarly for the noisy setting.

This work seems sufficiently different from the existing hyperspectral unmixing methods that are based on total variation penalty, and if correct, the theoretical analysis would be of interest to many members of the learning community who work with sparsity assumptions and graph-based learning. However, there are a few questions I would like to get clarified:
- The main concern about this method and its results is about the graph G. The paper claims that G does not have to accurately represent the similarity structure in the data, but that sounds counter-intuitive. Surely if the given graph is wrong (e.g. adversarial setting, complement graph), the method shouldn't work well? Is such adaptivity reflected in the derived bounds? G seems to only appear in the bound via diam(G), which is a function of G, but not a function of "correctness of G", which one might expect to see in such problem setting.
- How is/should be G derived for real data?
- At the end of section 3.2 - "Although, we note that combining groups of pixels in this manner can cause the images to appear at a lower resolution." Could the authors explain why this is true?
- What are other applications that the authors envision this work being used for, besides hyperspectral unmixing?
- In section 2.1, "For instance, in Hyperspectral Imaging we expect the composition of the ground to change by a few minerals when moving to an adjacent pixel." However, many hyperspectral unmixing methods assume the mineral composition vectors to sum to 1 (such that they can be interpreted as a probability/percentage vector). Are there ways to get around it?

**Time Spent Reviewing:**

3

---

> ### Author Response · Authors · 2021-08-09
> **Comment to Reviewer One**
>
> We provide answers to specific questions in the following bullet point format.
>
> * Affect of $G$ on learning
>     - See **Adaptitvity to unkown graph $G$ and edge sparsity** $\mathbf{s}^{\prime}$ in Comment to all Reviewers.
>
> * How is/should be G derived for real data?
>     - Information about $G$ can come from the problem setting e.g. geographic proximity between data centers for distributed computing or pixels for hyperspectral unmixing. More generally, different graphs can be considered and one chosen through  model selection e.g. simplest (smallest diameter) graph that fits the data. Estimating G from real data is outside of the scope of our present work. We believe providing theoretical guarantees when jointly estimating the graph and signals is an interesting direction for future research.
>
> * Combining groups of pixels causing images to appear at a lower resolution
>     - Coupling pixels introduces a correlation between them, making them likely to take *similar* value. Having 2x2 patches of similar valued pixels can appear as one larger pixel, and thus, over the entire image have the effect of reducing the resolution.
>
> * Applications besides hyerspectral unmixing
>     - The total variation penalty has seen applications within genomics see (S. Kim and E P. Xing, 2009), [10], and their citations. Meanwhile, the $\ell_2$ total variation penalty is routinely applied within decentralised problems, see [48,51] and references therein.  Our work lays the foundations for considering $\ell_1$ in this case, which, for instance, allows for sparse parameters and reduced communication cost between agents.
>     - Statistical Estimation of Correlated Genome Associations to a Quantitative Trait Network  Seyoung Kim, Eric P. Xing PLoS Genetics  2009
>
> * Coefficients as probabilities  / proportions in Hyperspectral imaging.
>     - The vector of coefficients can always be normalised to sum to one after their recovery. In this case, the coefficients will remain sparse (or approximately sparse) since the majority of minerals will make up a small percentage. A possible future direction of research is to consider approximately sparse vectors in, say, more general q-norms for $2 > q > 1$.

---

> ### Author Response · Authors · 2021-09-06
> **Follow up to Reviewer One**
>
> Dear Reviewer,
>
> Just a quick follow up regarding our response to your comments.
> Specifically, If you have any other outstanding concerns, do let us know as we can provide additional clarification.
>
> Otherwise, if you have no other concerns, we would kindly ask that you update your review in light of this.
>
> Best,
> Authors.

---

### Author Response · Authors · 2021-08-09
**Comment to all Revewers**

We thank the reviewers for their overall feedback. In this comment, we provide a response to questions common across the reviewers. Responses to each reviewer are then provided to address specific concerns.

$\textbf{Adaptivity to unknown graph G and edge sparsity } \mathbf{s}^{\prime}$
The sample complexities provided in Table 1 depend on the root sparsity $s$, graph topology (diameter and degree) $G$, and maximum  sparsity along edges $s^{\prime}$. When the algorithm *does not* depend $G$ or $s^{\prime}$ e.g. Total Variation Basis Pursuit Denoising (TVBP) on a star, then the *best case* over $G$ and $s^{\prime}$ can be taken. That is, TVBP on a star has total sample complexity $O(s + n^2 \text{Diam}(G) s^{\prime})$, so $G$ and $s^{\prime}$ which minimises the product $\text{Diam}(G)s^{\prime}$ can be taken. These details are now described within the manuscript.

$\textbf{Incorporating Graph knowledge}$ Following the above, when incorporating the graph information into the algorithm i.e. the complexities in Table 2, the graph $G$ used (within the algorithm) and the associated $s^{\prime}$, will be the one in the rate given. This case, and its the differences from Table 1, are now clarified within the manuscript.

---

### Decision · Program_Chairs · 2021-09-27

**Decision:**

Accept (Poster)

**Comment:**

While the reviewers have pointed out several concerns regarding the paper, there seems to a consensus that this paper has merit. Everyone is placing this paper as a borderline candidate.  I will strongly suggest that the authors implement the suggestions given by the reviewers in the subsequent versions.